# Coordination between ECM and cell-cell adhesion regulates the development of islet aggregation, architecture, and functional maturation

Wilma Tixi[1†], Maricela Maldonado[1,2†], Ya-Ting Chang[1†], Amy Chiu[1], Wilson Yeung[1], Nazia Parveen[1], Michael S Nelson[3], Ryan Hart[4], Shihao Wang[5], Wu Jih Hsu[5], Patrick Fueger[6], Janel L Kopp[5], Mark O Huising[4,7], Sangeeta Dhawan[1], Hung Ping Shih[1*]

[1]Department of Translational Research and Cellular Therapeutics, Arthur Riggs Diabetes and Metabolism Research Institute, Beckman Research Institute, City of Hope, Duarte, United States; [2]Department of Biomedical Engineering, College of Engineering, California State University, Long Beach, Long Beach, United States; [3]Light Microscopy Core, Beckman Research Institute, City of Hope, Duarte, United States; [4]Department of Neurobiology, Physiology and Behavior, University of California, Davis, Davis, United States; [5]Department of Cellular and Physiological Sciences, Life Sciences Institute, University of British Columbia, Vancouver, Canada; [6]Department of Molecular & Cellular Endocrinology, Arthur Riggs Diabetes and Metabolism Research Institute, Beckman Research Institute, City of Hope, Duarte, United States; [7]Department of Physiology and Membrane Biology, School of Medicine, University of California, Davis, Davis, United States

*For correspondence:
hshih@coh.org

†These authors contributed equally to this work

Competing interest: The authors declare that no competing interests exist.

**Abstract** Pancreatic islets are three-dimensional cell aggregates consisting of unique cellular composition, cell-to-cell contacts, and interactions with blood vessels. Cell aggregation is essential for islet endocrine function; however, it remains unclear how developing islets establish aggregation. By combining genetic animal models, imaging tools, and gene expression profiling, we demonstrate that islet aggregation is regulated by extracellular matrix signaling and cell-cell adhesion. Islet endocrine cell-specific inactivation of extracellular matrix receptor integrin β1 disrupted blood vessel interactions but promoted cell-cell adhesion and the formation of larger islets. In contrast, ablation of cell-cell adhesion molecule α-catenin promoted blood vessel interactions yet compromised islet clustering. Simultaneous removal of integrin β1 and α-catenin disrupts islet aggregation and the endocrine cell maturation process, demonstrating that establishment of islet aggregates is essential for functional maturation. Our study provides new insights into understanding the fundamental self-organizing mechanism for islet aggregation, architecture, and functional maturation.

## Editor's evaluation

This important study advances our knowledge regarding how islet endocrine cells interact with one another and with surrounding blood vessels during embryonic development and in adult mice. The evidence supporting this work is convincing, with complementary microscopy, functional and transcriptomic data. The authors' conclusions are supported by the data with sample size and quantitative analyses. These data should be of broad interest to islet biologists.

## Introduction

To control various physiological responses, groups of endocrine cells cluster together with vasculature, mesenchymal cells, and neuronal cells to form a highly organized 'mini-organ' in the pancreatic epithelium: the islet of Langerhans. The islet endocrine cells emerge from a domain of multipotent progenitors in the embryonic pancreatic epithelium. Initially, the multipotent pancreatic progenitors resolve into a pre-acinar domain, and a bipotent endocrine/ductal progenitor domain of the developing pancreas (*Larsen and Grapin-Botton, 2017*; *Romer and Sussel, 2015*; *Shih et al., 2013*). Endocrine cell progenitors are specified by the expression of the proendocrine gene *Neurogenin 3* (*Ngn3*) (*Gu et al., 2002*). Following endocrine cell specification, the Ngn3+ precursors undergo delamination (*Bechard et al., 2016*; *Gouzi et al., 2011*; *Rosenberg et al., 2010*). Delaminated endocrine precursors remain associated with the ducts as they migrate and join other endocrine cells and aggregate into immature islets (*Sharon et al., 2019a*). These immature islets form long interconnected cords along the ductal epithelium. Later the islet cords change morphology and undergo fission to form distinct spherical islets (*Jo et al., 2011*; *Seymour et al., 2004*; *Sznurkowska et al., 2020*). Throughout the differentiation and morphological changes, endocrine cells are intimately associated with the vasculature, which facilitates functional control of hormone release (*Reinert et al., 2014*). To coordinate the hormone release of endocrine cells within each islet, autocrine and paracrine interactions, as well as direct cell contacts, are required (*Campbell and Newgard, 2021*). These interactions are established by the aggregation process during development (*Adams and Blum, 2022*). Defects in the aggregation processes may lead to defects in endocrine cell development, and eventually, islet dysfunction. Supporting this notion, human pluripotent stem cell (hPSC)-derived β-cells acquire more functional maturity in vitro if clustered together (*Cottle et al., 2021*; *Nair et al., 2019*). However, the mechanism by which this process is regulated in vivo has largely remained elusive.

Self-organization of a group of cells into a higher-order, multicellular functional unit relies on an array of cell interactions regulated by cell-cell adhesions (*Fagotto, 2014*). For example, a calcium-dependent adhesion molecule, neural cell adhesion molecule (N-CAM), has been shown to be critical for murine endocrine cells to cluster (*Esni et al., 1999*). In rodents, islets form a stereotypical core-mantle architecture in which the insulin (Ins) secreting β-cells are located mainly in the central core, with glucagon (Gcg) secreting α-cells and somatostatin (Sst) secreting δ-cells localizing in the periphery to form a mantle (*Adams and Blum, 2022*; *Arrojo e Drigo et al., 2015*; *Kim et al., 2009*). In culture, dissociated rat endocrine cells can self-organize into the core-mantle architecture, suggesting that the islet structure is established by factors intrinsic to endocrine cells (*Halban et al., 1987*). N-CAM-deleted mice show abnormal islet architecture, with α-cells spread throughout the islet core and abnormal assembly of adherens junction proteins N- and E-cadherin (Ncad and Ecad) (*Esni et al., 1999*). In addition, ectopic expression of a dominant negative Ecad in mouse β-cells interferes with their clustering and islet formation (*Dahl et al., 1996*); and loss of *Ecad* results in abnormal blood glucose homeostasis (*Wakae-Takada et al., 2013*). Intracellularly, adherens junctions are stabilized by β-catenin (*Ctnnb1*) and α-catenin (*Ctnna1*) (*Jamora and Fuchs, 2002*). The function of *Ctnnb1* has been extensively studied in the developing pancreas (*Elghazi et al., 2012*; *Murtaugh et al., 2005*; *Rulifson et al., 2007*) and in hPSC-derived β-cells, because of its role in cell proliferation via Wnt signaling (*Sharon et al., 2019b*); yet genetic studies show that *Ctnnb1* is not essential for islet clustering and architecture (*Elghazi et al., 2012*; *Murtaugh et al., 2005*). In the developing pancreas, inactivation of *Ctnna1* results in defects in endocrine progenitor differentiation due to an expansion of pancreatic ductal progenitors (*Jimenez-Caliani et al., 2017*). Whether the process of islet clustering and architecture requires *Ctnna1* is unknown.

The regulation of cell-cell adhesion is often coupled with cell-extracellular matrix (ECM) interaction and integrin signaling (*Mui et al., 2016*; *Weber et al., 2011*). Upon ECM binding, integrins activate signal transduction pathways and regulate (1) cytoskeleton dynamics, (2) cell movement, (3) cell differentiation, and (4) functional maturation of epithelial cells, vasculature, and neurons (*Barros et al., 2011*; *Li et al., 2008*; *Scheppke et al., 2012*). During angiogenesis, ECM receptor integrin β1 (Itgb1) is required to control the localization of cell-cell adhesion molecule VE-cadherin and maintain cell-cell junction integrity (*Yamamoto et al., 2015*). In salivary glands and early embryonic pancreas, Itgb1-mediated ECM adhesions regulate Ecad to control cell clustering and regulate budding epithelial branching (*Shih et al., 2016*; *Wang et al., 2021*; *Wang et al., 2017*). Whether islet morphogenesis also requires ECM-integrin signaling remains to be investigated.

In the present study, we examine how loss of *Itgb1* and *Ctnna1* in Ngn3-expressing endocrine progenitors affects islet morphogenesis. We show that *Itgb1* and *Ctnna1* have collaborative roles during islet aggregation: ECM-Itgb1 signaling modulates islet aggregation by negatively regulating cell-cell adhesions; and α-catenin promotes endocrine cell clustering and modulates vascularization during islet development. We also provide evidence that the islet architecture is regulated by differential adhesion in endocrine subtypes.

## Results

### The morphogenesis and functional maturation of islet endocrine cells depend on ECM-Itgb1 signaling during development

To investigate the process of islet aggregation, embryonic day 15.5 (E15.5) pancreas was stained for blood vessel maker platelet endothelial cell adhesion molecule-1 (PECAM), ductal marker Spp1, and endocrine cell marker chromogranin A (ChrA). Confocal microscopy and three-dimensional (3D) whole-mount immunofluorescence (IF) imaging revealed that at E15.5 endocrine cells had started aggregating, and the vascular and ductal networks were already highly similar (*Figure 1—figure supplement 1A*). This close relation between the ducts and blood vessels at the stage prior to islet aggregation suggests that blood vessels could readily interact and provide instructive cues to regulate endocrine progenitor aggregation into islets. Blood vessels in islets produce ECM-rich basal membrane to support and organize endocrine cells (*Lammert et al., 2003*; *Roscioni et al., 2016*). Thus, we hypothesized that ECM signaling, provided by the vasculature, plays a key role in the initiation steps of islet aggregation. Supporting this notion, published work using single-cell gene expression profiling has shown that the ECM-Itgb1 signaling gene ontology (GO) categories are most significantly enriched in the Ngn3+ endocrine progenitors (*Bastidas-Ponce et al., 2019*). IF analysis validated that both the Ngn3+ endocrine progenitors and their progenies (eYFP+ cells in *Ngn3-Cre; Rosa26-eYFP* pancreas) expressed Itgb1 (*Figure 1A*). To investigate the role of Itgb1 during endocrine cell aggregation, we used the *Ngn3-Cre* allele to recombine the *Itgb1-flox* alleles in endocrine progenitors (*Ngn3-Cre; Itgb1^{f/f}*; hereafter *Itgb1^{ΔEndo/ΔEndo}*). Expression of Itgb1 was almost entirely absent throughout the islet endocrine cells in comparison to control littermates at 6 weeks of age (*Figure 1B and C*). We first examined whether inactivation of ECM-Itgb1 signaling affected vascular interaction. Electron microscope ultrastructure analysis revealed tight attachment in basal membranes between vascular and endocrine cells in control islets (*Figure 1—figure supplement 1B*). In contrast, in the *Itgb1^{ΔEndo/ΔEndo}* islets, the endocrine cells were detached from the basal membranes of blood vessels (*Figure 1—figure supplement 1C*, red arrows). Despite the vascular defects with Itgb1 loss, all four endocrine cell subtypes, α-, β-, δ-, and Ppy- (pancreatic polypeptide) cells, were present at similar proportions in control and *Itgb1^{ΔEndo/ΔEndo}* islets (*Figure 1D and E*; *Figure 1—figure supplement 1D–G*). In addition, the expression of β-cell lineage markers Nkx6.1 and Pdx1 were present in the Ins+ cells of control and *Itgb1^{ΔEndo/ΔEndo}* islets (*Figure 1F and G*). Although the populations of endocrine cell lineages were unaffected by the loss of *Itgb1*, morphometrical analysis revealed drastic morphological changes. In control islets, β-cells were organized in a central core and surrounded by α- and δ-cells in the periphery (*Figure 1D*). Conversely, in *Itgb1^{ΔEndo/ΔEndo}* mice, several individual islets appeared to agglomerate, resulting in larger islet sizes and aberrant distribution of α- and δ-cells in the central core (*Figure 1E and G*; *Figure 1—figure supplement 1I*). Thus, the inactivation of Itgb1 has a profound impact on islet morphogenesis, but not endocrine cell fate choice.

We next investigated whether the loss of *Itgb1* had any impact on the expression of genes related to islet function. Islets from 6-week-old control and *Itgb1^{ΔEndo/ΔEndo}* mice were isolated and subjected to RNA-sequencing (RNA-seq) and differential gene expression analysis. A comparison of gene expression profiles between *Itgb1^{ΔEndo/ΔEndo}* and control islets revealed 417 differentially expressed genes (DEGs) with false discovery rate (FDR)<0.05 and fold change (FC) >2 (*Supplementary file 1*), of which 332 were up-regulated and 85 were down-regulated. Gene-set enrichment analysis (GSEA) of DEGs between control and *Itgb1^{ΔEndo/ΔEndo}* islets showed up-regulation of pathways associated with hypoxia, epithelial mesenchymal transition, glycolysis, and inflammatory response (*Figure 1H*). The analysis of GO of DEGs related to hypoxia showed that the genes responsible for hypoxia response were up-regulated in *Itgb1^{ΔEndo/ΔEndo}* islets when compared to the control group (as shown in *Figure 1—figure supplement 1L*). The IF analysis indicated that the *Itgb1^{ΔEndo/ΔEndo}* islets at 2–3 months of age displayed

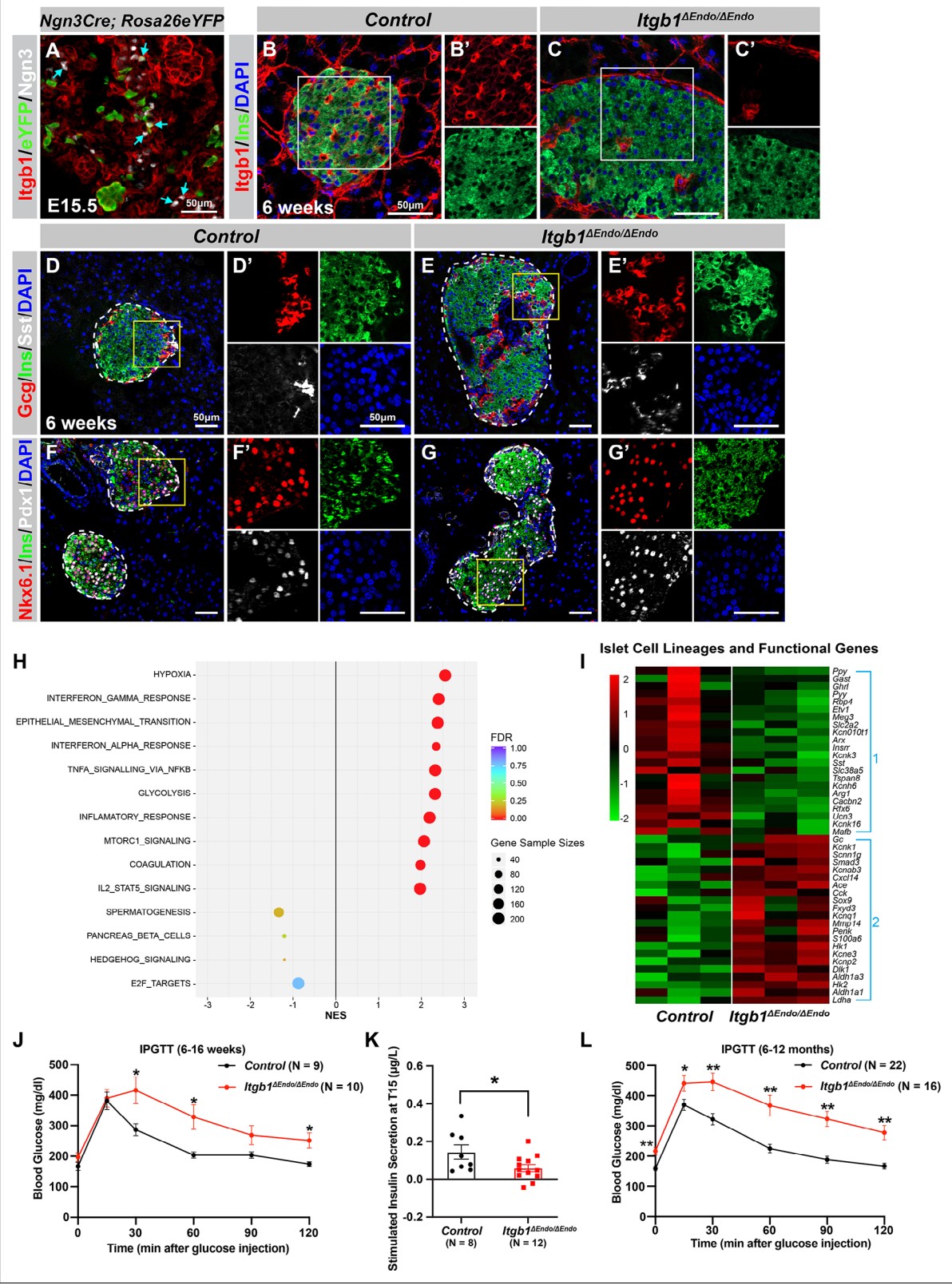

**Figure 1.** ECM-Itgb1 signaling in endocrine progenitors is required for normal islet morphology and function. (**A**) Immunofluorescence staining for Itgb1 and Ngn3 in pancreas sections of *Ngn3-Cre; Rosa26eYFP* reporter mice at E15.5. Cyan arrows indicate Ngn3[+] cells and lineage-traced endocrine cells expressing Itgb1. (**B–C**) Immunofluorescence staining for Itgb1, Ins, and DAPI on pancreas sections, demonstrating significant *Itgb1* deletion in an *Itgb1^ΔEndo/ΔEndo* islet at 6 weeks of age. Fields demarcated by white boxes in (**B–C**) are shown with individual color channels in (**B′–C′**) side panels.

*Figure 1 continued on next page*

*Figure 1 continued*

(**D–G**) Immunofluorescence staining for Gcg, Ins, Sst, Nkx6.1, Pdx1, and DAPI on pancreas sections of (**D, F**) control and (**E, G**) *Itgb1^{ΔEndo/ΔEndo}* mice at 6 weeks of age. Individual islets are outlined by dotted lines. Fields demarcated by yellow boxes in (**D–G**) are shown at higher magnification in (**D'–G'**) side panels. (**H**) Gene-set enrichment analysis of the differentially expressed genes, indicating top enriched pathways for control versus *Itgb1^{ΔEndo/ΔEndo}* islets at 6 weeks of age. Gene sample sizes and FDRs are indicated by the size and color of the dots. (**I**) Heatmap demonstrating down-regulation (Group 1) and up-regulation (Group 2) of functional maturation genes in *Itgb1^{ΔEndo/ΔEndo}* islets compared to controls. (**J–L**) Intraperitoneal glucose tolerance test (**J, L**) and stimulated insulin secretion at T15 (**K**) on 6- to 16-week-old (**J, K**) and (**L**) 6- to 12-month-old control and *Itgb1^{ΔEndo/ΔEndo}* mice. Ngn3, neurogenin 3; Gcg, glucagon; Ins, insulin; DAPI, 4′,6-diamidino-2-phenylindole; Sst, somatostatin; IPGTT, intraperitoneal glucose tolerance test; FDR, false discovery rate; E15.5, embryonic day 15.5. Data are shown as mean ± SEM. *p<0.05 and **p<0.01.

The online version of this article includes the following source data and figure supplement(s) for figure 1:

**Source data 1.** Related to *Figure 1J*.

**Source data 2.** Related to *Figure 1K*.

**Source data 3.** Related to *Figure 1L*.

**Figure supplement 1.** Deletion of *Itgb1* in endocrine progenitors results in abnormal interactions with blood vessels and decreased expression of functional maturation markers within islets.

**Figure supplement 1—source data 1.** Related to *Figure 1—figure supplement 1D*.

**Figure supplement 1—source data 2.** Related to *Figure 1—figure supplement 1E*.

**Figure supplement 1—source data 3.** Related to *Figure 1—figure supplement 1F*.

**Figure supplement 1—source data 4.** Related to *Figure 1—figure supplement 1G*.

**Figure supplement 1—source data 5.** Related to *Figure 1—figure supplement 1J*.

**Figure supplement 1—source data 6.** Related to *Figure 1—figure supplement 1K*.

**Figure supplement 1—source data 7.** Related to *Figure 1—figure supplement 1O*.

**Figure supplement 2.** Deletion of *Itgb1* in endocrine progenitors results in abnormal interactions with blood vessels and decreased expression of functional maturation markers within islets.

**Figure supplement 2—source data 1.** Related to *Figure 1—figure supplement 2A*.

a significant reduction in blood vessels (as shown in *Figure 1—figure supplement 1M–O*), which suggests that the lack of vascular interaction could lead to hypoxia responses. On the other hand, the down-regulated genes were associated with spermatogenesis, pancreas β-cells, and E2F targets (*Figure 1H*). GO analysis of DEGs in the pancreas β-cells category further revealed that functional maturation genes for β-cells (e.g., *Slc2a2* [*Glut2*], *Ucn3*, and *Mafa*) and for α-cells (e.g., *Arx*, *Mafb*, and *Etv1*), were drastically down-regulated in the *Itgb1^{ΔEndo/ΔEndo}* islets (*Figure 1I*, Group 1; *Figure 1—figure supplement 2A*). IF analysis also validated the reduced expression of Mafa, Ucn3, and Glut2 in β-cells of *Itgb1^{ΔEndo/Δendo}* islets (*Figure 1—figure supplement 2B–E*). Furthermore, several 'disallowed' genes for β-cells, including *Ldha1*, *Aldh1a3*, and *Hk1* (*Lemaire et al., 2016*), were significantly up-regulated in the *Itgb1^{ΔEndo/ΔEndo}* islets (*Figure 1I*, Group 2; *Figure 1—figure supplement 2A*). Hence, it is likely that the functional maturation of endocrine cells within *Itgb1^{ΔEndo/ΔEndo}* islets is compromised. This notion is supported by the observation that, by 6 weeks of age, these mice exhibit glucose intolerance and reduced insulin secretion in response to glucose stimulation (*Figure 1J–K*). The impairment in glycemic regulation further deteriorates in mice older than 6 months (*Figure 1L*). Together, these observations indicate that ECM-integrin interactions are critical for islet vascularization, morphogenesis, and the functional maturation of endocrine cells.

## ECM signaling regulates endocrine progenitor cell migration and cell-cell adhesion to control islet aggregation during development

Our initial analysis of the pancreatic sections from *Itgb1^{ΔEndo/ΔEndo}* mice revealed larger islet sizes (*Figure 1—figure supplement 1J*). Consistently, the islets isolated from *Itgb1^{ΔEndo/ΔEndo}* mice appeared larger in size as well (*Figure 2A–C*); however, the total islet area was similar between control and *Itgb1^{ΔEndo/ΔEndo}* mice (*Figure 2D*). The increased size was mainly attributed to a greater number of large islets (>50 µm² × 10³, as shown in *Figure 1—figure supplements 1K and 2E*). Therefore, the enlarged islet size is likely a result of a morphological change independent of the expansion of total endocrine cells. Furthermore, in the control pancreata, the islets were located near the ducts, but contained a very small number of ductal cells (*Figure 2F and H*; *Figure 2—figure supplement 1A*). In contrast,

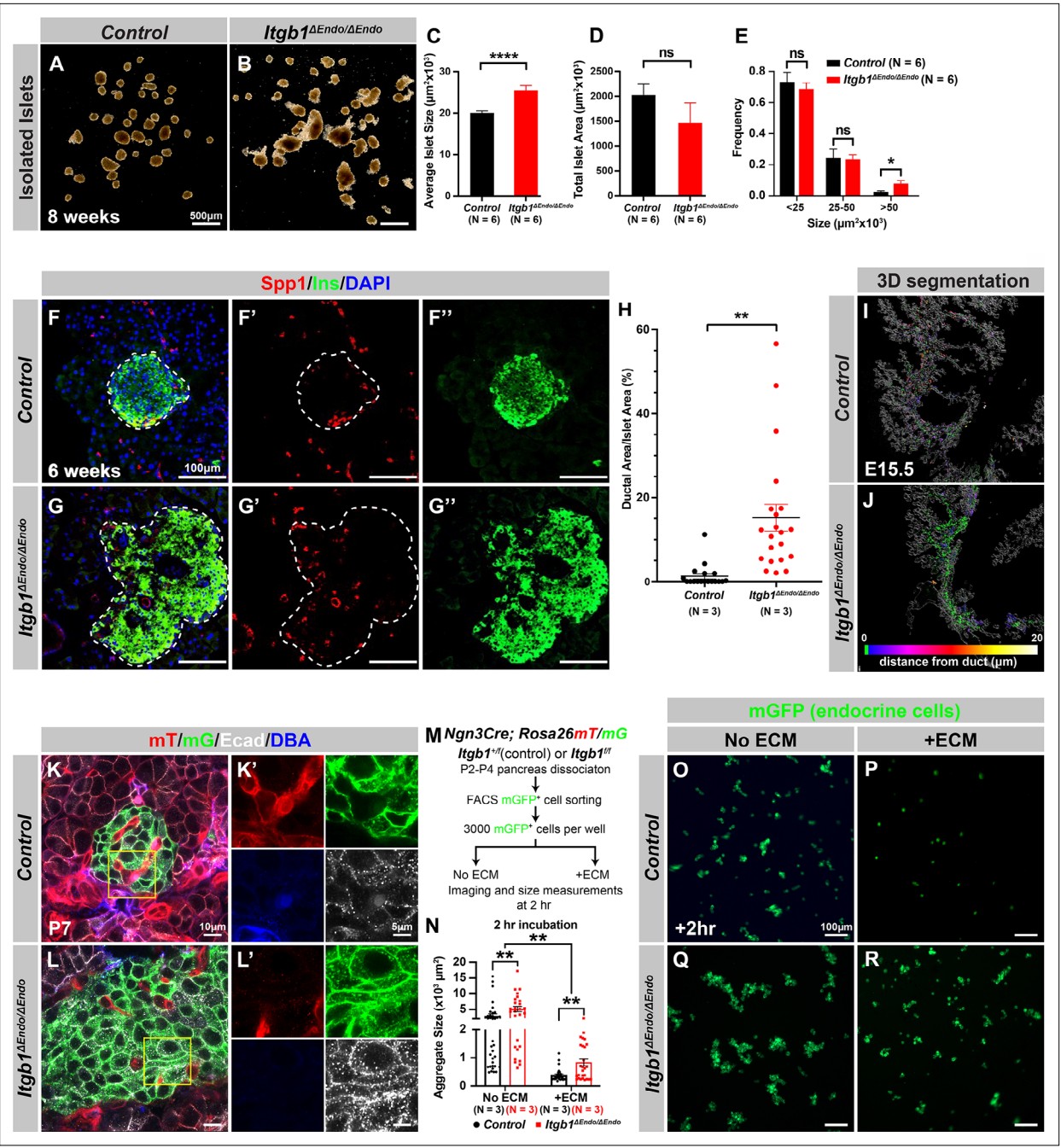

**Figure 2.** Itgb1 signaling is required for proper endocrine progenitor aggregation. (**A–B**) Brightfield images of isolated islets from control and *Itgb1^ΔEndo/ΔEndo* mice at 8 weeks of age. (**C–E**) Quantification of (**C**) average islet size, (**D**) total islet area, and (**E**) islet size distribution based on morphometric analysis of 8-week-old control and *Itgb1^ΔEndo/ΔEndo* isolated islets. (**F–G**) Immunofluorescence staining for Spp1, Ins, and DAPI on 6-week-old control and *Itgb1^ΔEndo/ΔEndo* pancreas sections. Islets are outlined by white dotted lines. (**H**) The ratio of ductal cell areas within the islets is quantified based on Spp1 and ChrA staining of pancreas sections from 6-week-old control and *Itgb1^ΔEndo/ΔEndo* mice. (**I–J**) Three-dimensional (3D) segmentation analysis on whole-mount immunofluorescence images of E15.5 control and *Itgb1^ΔEndo/ΔEndo* pancreata stained for ChrA and Spp1 (white colored) reveals that the endocrine cells in the *Itgb1^ΔEndo/ΔEndo* pancreas are mostly retained within ducts, and the distance between endocrine cells and ducts is represented by a color scale. (**K–L**) Airyscan super-resolution images depict immunofluorescence staining for Ecad and ductal marker DBA in P7 mT/mG reporter mice showing up-regulation of Ecad in the *Itgb1^ΔEndo/ΔEndo* pancreas. The dot-like Ecad expression pattern in super-resolution imaging represents adherens junction structures. The yellow boxes in (**K–L**) demarcate fields shown at higher magnification in (**K'–L'**) side panels. (**M**) The schematic depicts the experimental design for the in vitro aggregation assay. (**N**) Aggregate size is quantified after a 2 hr incubation with or without extracellular matrix (ECM). (**O–R**) Fluorescence images of mGFP+ endocrine progenitor aggregates after 2 hr of culture show that mGFP+ control endocrine progenitors clustered (**O**) without ECM but remained single cells (**P**) with ECM. Alternatively, *Itgb1^ΔEndo/ΔEndo* endocrine progenitors clustered (**Q**) without ECM or (**R**) with ECM.

*Figure 2 continued on next page*

*Figure 2 continued*

Spp1, secreted phosphoprotein 1; Ins, insulin; DAPI, 4′,6-diamidino-2-phenylindole; Ecad, E-cadherin; DBA, *Dolichos biflorus* agglutinin; P7, postnatal day 7; E15.5, embryonic day 15.5. Data are shown as mean ± SEM. *p<0.05, **p<0.01, and ****p<0.0001.

The online version of this article includes the following source data and figure supplement(s) for figure 2:

**Source data 1.** Related to *Figure 2C*.

**Source data 2.** Related to *Figure 2D*.

**Source data 3.** Related to *Figure 2E*.

**Source data 4.** Related to *Figure 2H*.

**Source data 5.** Related to *Figure 2N*.

**Figure supplement 1.** Itgb1 is required for normal islet architecture.

**Figure supplement 1—source data 1.** Related to *Figure 2—figure supplement 1I*.

**Figure supplement 2.** Itgb1 controls cytoskeleton regulation and organization in islets.

**Figure supplement 2—source data 1.** Related to *Figure 2—figure supplement 2B*.

the islets in the *Itgb1*$^{\Delta Endo/\Delta Endo}$ pancreata contained a substantial number of ductal cells (*Figure 2G and H*; *Figure 2—figure supplement 1B*). As the endocrine progenitors arise and migrate away from the ducts, we hypothesized that there might be deficiencies in this process for Itgb1-deficient endocrine cells. To investigate this, we conducted segmentation analysis of 3D whole-mount IF on E15.5 pancreata, the stage at which the majority of endocrine progenitor migration occurs (*Rosenberg et al., 2010*). Our results showed that most of the Itgb1-deleted endocrine progenitor cell aggregates had a greater tendency to cluster within the ductal cords during development (*Figure 2I and J* and *Video 1*; the green color represents zero distance from ducts). These findings imply that there might be deficiencies in both endocrine cell migration and premature islet aggregation in *Itgb1*-deleted endocrine cells. To further confirm this, we conducted live imaging analysis on pancreas explants obtained from E16.5 *Ngn3-Cre; Rosa26mT/mG; Itgb1*$^{+/+}$ (control) and *Ngn3-Cre; Rosa26mT/mG; Itgb1*$^{f/f}$ (*Itgb1*$^{\Delta Endo/\Delta Endo}$) mice. We tracked the movement and aggregation of Ngn3-progeny (mGFP$^+$) endocrine progenitor cells in explant cultures for up to 3 days (*Figure 2—figure supplement 1C and D*; *Videos 2 and 3*). Our observations showed that in contrast to the controls, where endocrine progenitor cells migrated and aggregated outside of ductal cords, the mGFP$^+$ cells in *Itgb1*$^{\Delta Endo/\Delta Endo}$ explants readily aggregated before moving away from the cords (*Figure 2—figure supplement 1D*, white arrows). In contrast to our findings, previous studies using *Ins-Cre* or *RIP-Cre* to ablate *Itgb1* after endocrine progenitors have differentiated

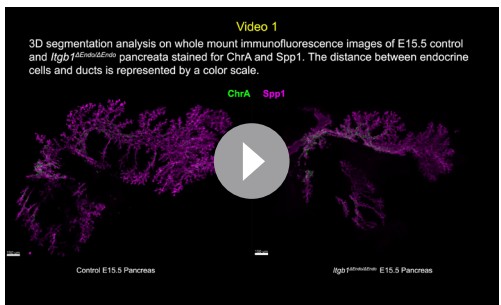

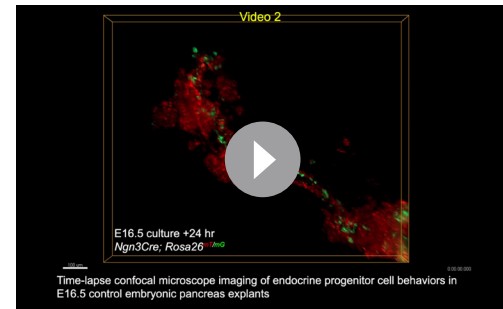

**Video 1.** Three-dimensional (3D) segmentation analysis on whole-mount immunofluorescence images of embryonic day 15.5 (E15.5) control and *Itgb1*$^{\Delta Endo/\Delta Endo}$ pancreata. This file presents a 3D segmentation analysis based on the chromogranin A (ChrA) and secreted phosphoprotein 1 (Spp1) staining, revealing islet clusters and pancreatic ducts. It includes movie clips showcasing the 3D segmentation from various viewing angles. A color scale illustrates the distance between endocrine cells and ducts, with cooler colors indicating shorter distances.

https://elifesciences.org/articles/90006/figures#video1

**Video 2.** Time-lapse confocal microscope imaging of endocrine progenitor cell behaviors in embryonic day 15.5 (E16.5) control (*Ngn3Cre; Rosa26*$^{mT/mG}$) embryonic pancreas explants. This file contains a time-lapse video showcasing the dynamic behaviors of endocrine progenitor cells. The green cell clusters visible in the footage represent *Ngn3Cre*-expressing progenies, which are the islet endocrine cells. The 8 s video clip encapsulates the changes observed over a 24 hr period.

https://elifesciences.org/articles/90006/figures#video2

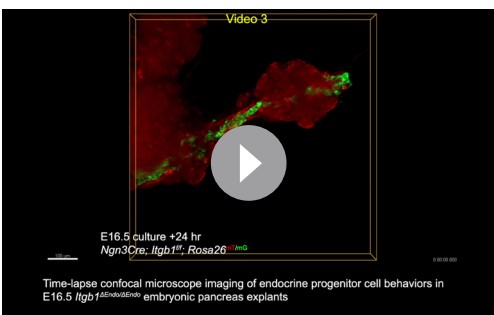

**Video 3.** Time-lapse confocal microscope imaging of endocrine progenitor cell behaviors in embryonic day 15.5 (E16.5) *Itgb1*$^{\Delta Endo/\Delta Endo}$ (*Ngn3Cre; Itgb1*$^{f/f}$*; Rosa26*$^{mT/mG}$) embryonic pancreas explants. This file includes a time-lapse video illustrating the behavior dynamics of *Itgb1*-deleted endocrine progenitor cells. The green cell clusters depicted in the footage represent *Ngn3Cre*-expressing progenies deficient in Itgb1. Changes observed over a 24 hr period are captured in an 8 s video clip. Notably, the *Itgb1*-deleted endocrine progenitor cells prematurely cluster into larger aggregates and maintain a close association with ductal epithelial cords.

https://elifesciences.org/articles/90006/figures#video3

into insulin-producing cells did not report any observable islet aggregation or migration defects (*Diaferia et al., 2013*; *Win et al., 2020*). However, *Ngn3-Cre*-mediated *Itgb1* deletion would occur much earlier in embryonic stages (*Figure 1A*), suggesting that Itgb1 signaling regulates the migration and aggregation of progenitor cells in the endocrine lineage during the islet developmental process.

To further investigate this notion, we sought to inactivate Itgb1 only after the endocrine cells had fully differentiated and matured. Urocortin 3 (Ucn3) is a late β-cell-specific maturity marker in mouse islets that is expressed after *Ins1* and *Ins2* and starts perinatally. Most β-cells express Ucn3 by 3 weeks of age (*Blum et al., 2014*; *van der Meulen and Huising, 2014*). The expression pattern of Cre recombinase under the control of the *Ucn3* promoter aligns with this timeline (*van der Meulen et al., 2017*). Therefore, *Ucn3-Cre*-mediated deletion of *Itgb1* is not expected to occur until islet development has completed in the presence of functional Itgb1 alleles and normal Itgb1 expression. To test whether Itgb1 signaling also maintains islet cell aggregation after β-cell maturation and islet formation, we crossed *Ucn3-Cre* with *Itgb1*$^{f/f}$ mice to inactivate Itgb1 exclusively in mature β-cells (referred to as *Itgb1*$^{\Delta Mature\beta/\Delta Mature\beta}$). Our analysis revealed that, similar to *Itgb1*$^{\Delta Endo/\Delta Endo}$ mice, *Itgb1*$^{\Delta Mature\beta/\Delta Mature\beta}$ mice exhibited vasculature exclusion from islets (*Figure 2—figure supplement 1E' and F'*), but islet size remained similar to control mice (*Figure 2—figure supplement 1E–1I*). These results suggest that Itgb1 signaling specifically regulates islet cell aggregation during pancreatic development.

During the early stages of pancreas development, the expression of Itgb1 in pancreatic progenitor cells has been identified as a crucial factor in controlling pancreas branching morphogenesis through its regulation of cell-cell adhesion (*Shih et al., 2016*). As the process of endocrine progenitor cell differentiation and islet formation commences, cell-cell adhesion is typically down-regulated to facilitate the migration and interaction of endocrine cells with non-endocrine cells (*Gouzi et al., 2011*). Given the established role of Itgb1 in regulating cell-cell adhesion in the context of pancreas development, we were prompted to investigate whether Itgb1 signaling also plays a role in instructing endocrine progenitor cells to modulate their adhesion behavior and thereby control the islet aggregation process. To address this question, we took Airyscan super-resolution images of Ecad IF on endocrine cells in *Ngn3-Cre; Rosa26mT/mG; Itgb1*$^{f/f}$ islets at P7, since islet aggregation is complete by the neonatal stages (*Sznurkowska et al., 2020*). The Airyscan images provide sufficient resolution to show adherens junctions between cells by visualizing Ecad$^+$ clusters (*Gonschior et al., 2020*; white dots in *Figure 2K and L*). The lineage-traced endocrine cells (mGFP$^+$) in control islets displayed low amounts of Ecad$^+$ clusters (*Figure 2K'*). However, the amount of Ecad$^+$ clusters was drastically increased in mutant islets (*Figure 2L'*), suggesting that Itgb1 signaling negatively regulates cell adhesion in developing islets. To further test whether the increased cell adhesiveness is responsible for the increased islet size when Itgb1 signaling is disrupted, we performed an in vitro cell aggregation assay. Live mGFP$^+$ endocrine cells, from neonatal (P2-P4) *Ngn3-Cre; Rosa26mT/mG; Itgb1*$^{+/f}$ or *Ngn3-Cre; Rosa26mT/mG; Itgb1*$^{f/f}$ pancreata, were completely dissociated, then collected by fluorescence-activated cell sorting (FACS). 3000 mGFP$^+$ cells were allowed to aggregate in wells of a 96-well plate that were coated with bovine serum albumin (BSA) (no ECM) or Matrigel and fibronectin (+ECM) (*Figure 2M*). Within 2 hr of cell seeding, the control mGFP$^+$ endocrine cells rapidly aggregated in wells with no ECM coating, but formed no aggregates in the ECM-coated wells (*Figure 2N–P*). Within the same amount of time,

the mutant mGFP⁺ endocrine cells clustered into larger aggregates in wells without ECM coating and maintained aggregation in the ECM-coated wells (*Figure 2N, Q and R*). After 48 hr, the mGFP⁺ cells isolated from both the controls and mutants progressively clustered into larger aggregates in wells without ECM coating (*Figure 2—figure supplement 2A, C, E*). In contrast, in the ECM-coated wells, only the mutant mGFP⁺ cells clustered, while control mGFP⁺ cells remained unaggregated (*Figure 2—figure supplement 2D and F*). Together, these findings demonstrate that ECM-Itgb1 signaling negatively regulates islet aggregation. As integrins play a crucial role in linking the actin cytoskeleton, we examined whether the inactivation of Itgb1 in endocrine progenitors affects the organization of the actin cytoskeleton in islet cells. In *Itgb1^{ΔEndo/ΔEndo}* islets, we observed pronounced foci of condensed F-actin between the endocrine cells (*Figure 2—figure supplement 2G–J*), indicating increased cell-cell adhesion. These findings suggest that the loss of Itgb1 in endocrine cells leads to alterations in the actin cytoskeleton, resulting in enhanced cell-cell adhesion within the islet microenvironment. The concentrated F-actin foci between the endocrine cells indicate a potential mechanism through which Itgb1 regulates cell-cell interactions and influences islet development and aggregation.

## α-Catenin promotes endocrine cell aggregation and regulates vascularization during islet development

Our study revealed that Itgb1 plays a critical role in islet development by regulating endocrine cell-cell adhesion. However, the mechanism by which cell-cell adhesion regulates islet formation remains unclear. Although Ecad and Ncad-mediated adherens junctions are believed to be the primary cell-cell adhesion mechanisms linking islet cells into aggregates, our findings and other studies have shown that deletions of Ecad or Ncad alone do not affect islet aggregation, suggesting functional redundancy between these cell adhesion molecules.

α-Catenin (Ctnna1) is an essential hub protein linking Ecad-mediated adherens junctions to the cytoskeleton and has been demonstrated to be indispensable for the formation of adherens junctions (*Jamora and Fuchs, 2002*). Therefore, to investigate the requirement of cell adhesion in islet formation, we conditionally ablated Ctnna1 in endocrine progenitors by generating *Ngn3-Cre; Ctnna1^{f/f}* mice (hereafter referred to as *Ctnna1^{ΔEndo/ΔEndo}*). To determine whether *Ctnna1* deletion affects islet development, ChrA staining on pancreas sections from 8-week-old mice were imaged, and the total relative islet area was measured (*Figure 3A and B*). Compared to their wild-type littermates, islet area (relative to the total pancreas area) in *Ctnna1^{ΔEndo/ΔEndo}* mice was unchanged (*Figure 3C*), but there was a significant reduction in average islet size (*Figure 3D*). The proportion of larger islets was significantly reduced, while the proportion of small islets was increased (*Figure 3—figure supplement 1A–C*), suggesting that *Ctnna1* is required for proper islet aggregation, but not for the development of total islet cell mass. In *Ctnna1^{ΔEndo/ΔEndo}* islets, we observe diminished Ecad expression and a decrease in the density of F-actin at cell junctions. These findings imply a disruption of cell junctions and a reduction in cell adhesion among Ctnna1-deficient endocrine cells (*Figure 3E and F*).

To investigate how *Ctnna1* deletion impacts islet development, we compared transcriptional profiling of *Ctnna1^{ΔEndo/ΔEndo}* and control islets at 8 weeks. A comparison of gene expression profiles between *Ctnna1^{ΔEndo/ΔEndo}* and control islets revealed significant differences in the expression of 319 genes with FDR <0.05 and FC >2, of which 245 were up-regulated and 74 were down-regulated (*Supplementary file 2*). Among all GO categories, one of the most up-regulated groups in *Ctnna1^{ΔEndo/ΔEndo}* islets is genes associated with ECM signaling and blood vessel interactions (*Figure 3G*; *Figure 3—figure supplement 1D*). These include metalloproteases, vascular adhesion, collagens, integrins, laminins, glycoproteins, and thrombospondins (*Figure 3G*). These findings suggest that *Ctnna1^{ΔEndo/ΔEndo}* islets have augmented ECM signaling and blood vessel interaction. IF for blood vessels confirmed an increase in blood vessel density in *Ctnna1^{ΔEndo/ΔEndo}* islets at 8 weeks (*Figure 3H–J*). The increase in islet vasculature was observed as early as P14 (*Figure 3K–L*). These findings demonstrated that the inactivation of *Ctnna1* in endocrine progenitor cells led to a decrease of endocrine cell-cell adhesion and an increase in blood vessel interaction.

Alongside the increased interaction with blood vessels, we observed abnormal organization of F-actin fibers within *Ctnna1^{ΔEndo/ΔEndo}* islets (*Figure 3E and F*; *Figure 3—figure supplement 1E and F*). These alterations in F-actin fiber organization were evident as early as P14 in the *Ctnna1^{ΔEndo/ΔEndo}* islets, concurrent with the onset of enhanced interaction with blood vessels (*Figure 3—figure supplement 1G and H*). These findings demonstrate that the loss of Ctnna1 compromises endocrine cell

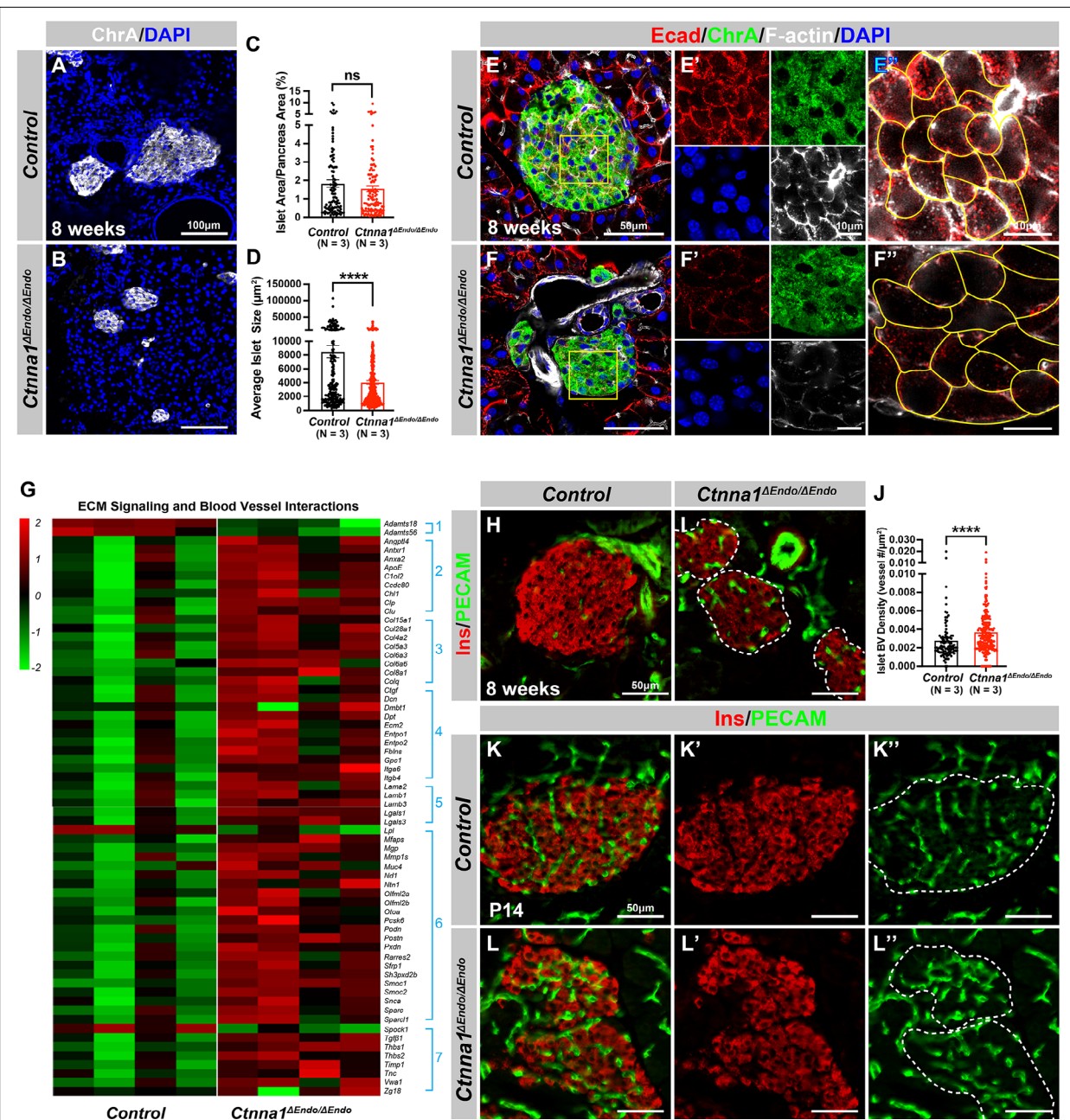

**Figure 3.** Ctnna1-mediated cell-cell adhesion is required for normal islet morphology and endocrine cell aggregation. (**A–B**) Immunofluorescence staining for ChrA and DAPI in 8-week-old pancreas sections showing smaller islet sizes in *Ctnna1^(ΔEndo/ΔEndo)^* mice. (**C–D**) Quantification of (**C**) the percentage of islet area in pancreas area and (**D**) average islet size based on ChrA staining on 8-week-old control and *Ctnna1^(ΔEndo/ΔEndo)^* pancreas sections. (**E–F**) Airyscan images of immunofluorescence staining for Ecad, ChrA, F-actin, and DAPI for control and *Ctnna1^(ΔEndo/ΔEndo)^* pancreas sections. Fields demarcated by yellow boxes are shown at higher magnification with individual color channels in (**E'–F'**) middle panels, showing reduced expression of Ecad and F-actin in *Ctnna1^(ΔEndo/ΔEndo)^* islets. Individual endocrine cell shape is delineated by yellow lines in (**E''–F''**), demonstrating the enlarged endocrine cell sizes in the islets of *Ctnna1^(ΔEndo/ΔEndo)^* mice. (**G**) Heatmap showing genes for extracellular matrix (ECM) signaling and blood vessel interactions are up-regulated in *Ctnna1^(ΔEndo/ΔEndo)^* islets at 8 weeks of age. Groups 1–7 represent gene ontologies of: (1) metalloproteases, (2) vascular adhesion, (3) collagens, (4) integrins, (5) laminins, (6) glycoproteins, and (7) thrombospondins. (**H–I**) Immunofluorescence staining of Ins and PECAM on 8-week-old control and *Ctnna1^(ΔEndo/ΔEndo)^* pancreas sections. (**J**) Quantification of islet blood vessel density based on Ins and PECAM staining on 8-week-old control and *Ctnna1^(ΔEndo/ΔEndo)^* pancreas sections. (**K–L**) Immunofluorescence staining of Ins and PECAM on P14 control and *Ctnna1^(ΔEndo/ΔEndo)^* pancreas sections showing increased vasculature in *Ctnna1^(ΔEndo/ΔEndo)^* islets. Islet area is outlined by dotted lines. ChrA, chromogranin A; DAPI, 4',6-diamidino-2-phenylindole; Ecad, E-cadherin; Ins, insulin; PECAM, platelet endothelial cell adhesion molecule-1; P14, postnatal day 14. Data are shown as mean ± SEM. ****p<0.0001.

The online version of this article includes the following source data and figure supplement(s) for figure 3:

*Figure 3 continued on next page*

*Figure 3 continued*

**Source data 1.** Related to *Figure 3C*.

**Source data 2.** Related to *Figure 3D*.

**Source data 3.** Related to *Figure 3J*.

**Figure supplement 1.** *Ctnna1*-deleted pancreatic islet endocrine cells display cytoskeleton anomalies.

**Figure supplement 1—source data 1.** Related to *Figure 3—figure supplement 1C*.

---

aggregation and perturbs the organization of the actin cytoskeleton and cell-cell adhesion. Together, these findings suggest that the balance between cell-cell adhesion and blood vessel interaction is crucial for islet cell aggregation during development (*Figure 3—figure supplement 1I*).

## Inactivation of Ctnna1 in differentiating endocrine cells affects the assembly and maintenance of non-β-endocrine cells in islets

To further investigate the impact of Ctnna1-dependent islet cell aggregation on the islet development process, we compared the gene expression profiles of $Ctnna1^{\Delta Endo/\Delta Endo}$ and control islets. Our analysis revealed significant misregulation of genes crucial for endocrine cell functions. Specifically, we observed a down-regulation of *Slc27a2*, *Tmsb15b1*, and *Etv1* expression, which are genes required for glycemic control and insulin secretion (*Figure 4A* Group 1; *Figure 4—figure supplement 1A*,). Conversely, we found an up-regulation of *Hk1*, *Ldha*, *Aldh1a1*, and *Cxcl14* expression, which are genes known to be 'disallowed' in mature functional β-cells (*Lemaire et al., 2016*; *Figure 4A* Group 2; *Figure 4—figure supplement 1A*). These findings suggest that the function of β-cells may be negatively impacted in $Ctnna1^{\Delta Endo/\Delta Endo}$ mice.

To support this notion, we conducted glucose tolerance tests, which revealed mild glucose intolerance in $Ctnna1^{\Delta Endo/\Delta Endo}$ mice at 8 weeks (*Figure 4B*). However, the quantification of insulin content in serum after glucose challenge did not reveal any noticeable insulin secretion defect in the $Ctnna1^{\Delta Endo/\Delta Endo}$ mice (*Figure 4C*). We reasoned that the conventional serum insulin measurement may not reflect the changes in β-cell function, given that the $Ctnna1^{\Delta Endo/\Delta Endo}$ mice exhibit only mild glucose intolerance. Therefore, we performed quantification of insulin secretion in vivo using hyperglycemic clamps, which is a more sensitive detection method. In this assay, we observed a trend toward lower overall insulin secretion in the $Ctnna1^{\Delta Endo/\Delta Endo}$ mice at 8 weeks (*Figure 4—figure supplement 1B and C*), providing additional evidence that Ctnna1-dependent islet cell aggregation has an impact on β-cell function. In addition, the glucose intolerance phenotype became even more pronounced at 1 year of age, as evidenced by the glucose tolerance test (*Figure 4D*). These findings indicate that Ctnna1-dependent islet cell aggregation is crucial for the proper development of β-cells and their function in maintaining glucose homeostasis.

In addition to suggested functional defects in β-cells, we found that genes for non-β-endocrine cell lineages, including α-cell genes *Gcg*, *MafB*, *Arx*, *Peg10,* and *Etv1,* and the δ-cell gene *Sst*, were significantly down-regulated in $Ctnna1^{\Delta Endo/\Delta Endo}$ islets (*Figure 4A*; *Figure 4—figure supplement 1A*), suggesting the requirement of *Ctnna1* for the presence of non-β-endocrine cells in islets. Interestingly, $Ctnna1^{\Delta Endo/\Delta Endo}$ islets did not show a reduced area of β-cells, as β-cells were still loosely organized in the core of islets (*Figure 4E–G*). However, there was a severe reduction of α-cell and δ-cell populations in the *Ctnna1*-deleted islets (*Figure 4H*; *Figure 4—figure supplement 1D–F*). To determine whether the loss of non-β-endocrine cells was the result of differentiation defects during development, we examined the population of α- and β-cells in $Ctnna1^{\Delta Endo/\Delta Endo}$ islets at P0 and P7. At these ages, the islets of $Ctnna1^{\Delta Endo/\Delta Endo}$ mice exhibited normal endocrine cell mass, Gcg$^+$ α-cell and Sst$^+$ δ-cell areas (*Figure 4I–K*; *Figure 4—figure supplement 1G–J*), demonstrating that the loss of α- and δ-cell was not due to differentiation defects prior to these early stages. However, the detachment of α-cells from the mantle of $Ctnna1^{\Delta Endo/\Delta Endo}$ islets was observed by P7. By this timepoint, α-cells had formed tight interactions with other endocrine cells in control islets (*Figure 4—figure supplement 1I and K*), but *Ctnna1*-deficient α-cells detached from other cell types (*Figure 4—figure supplement 1J and L*). To investigate whether cell detachment promotes cell death, thus resulting in α-cell loss, TUNEL analysis on P7 $Ctnna1^{\Delta Endo/\Delta Endo}$ islets was conducted to analyze cell apoptosis. Compared to the control, $Ctnna1^{\Delta Endo/\Delta Endo}$ islets display three times more TUNEL$^+$ apoptotic α-cells (*Figure 4L–N*), indicating that *Ctnna1* is required for α-cell survival. Thus, the loss of the α-cells in $Ctnna1^{\Delta Endo/\Delta Endo}$

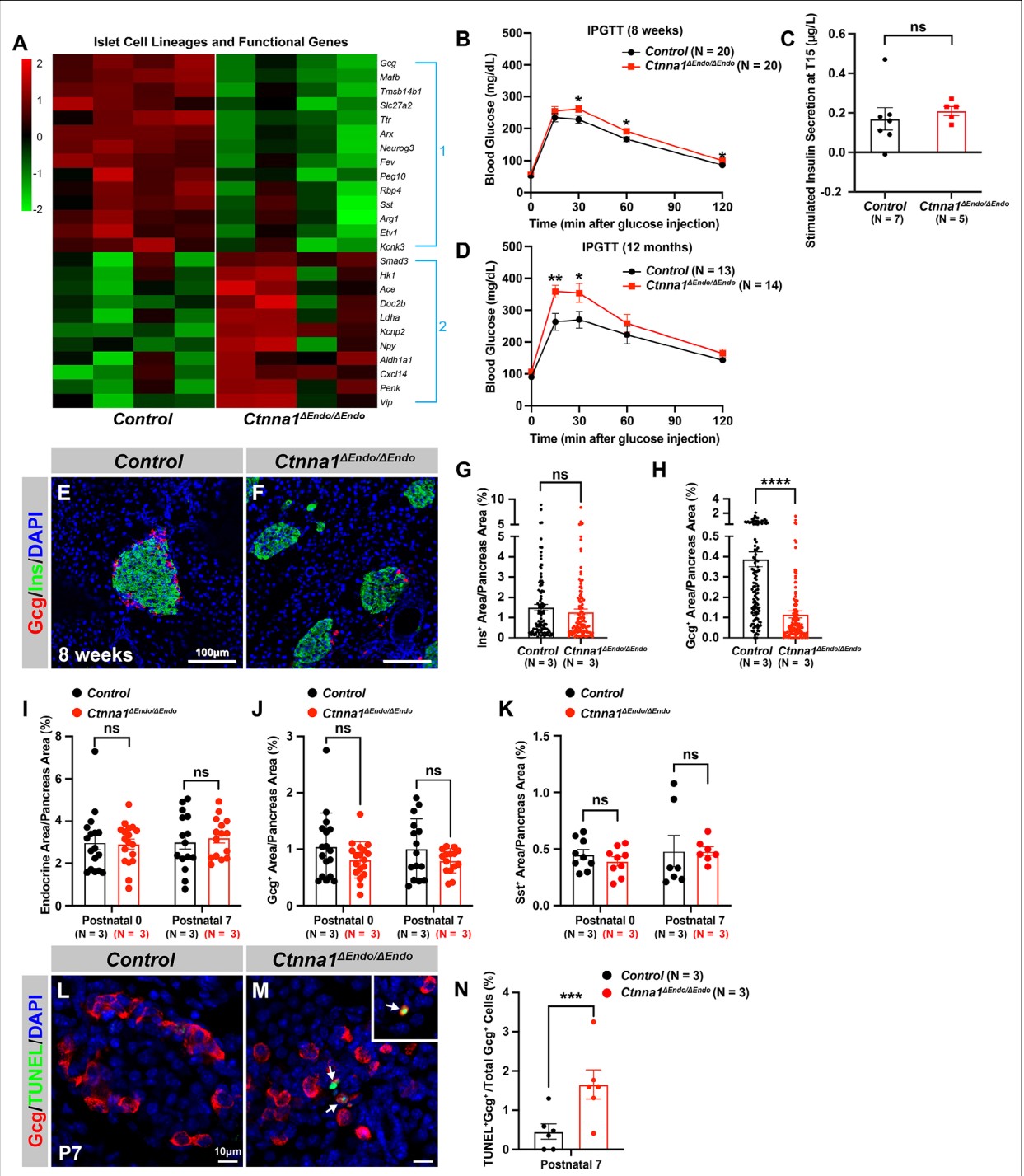

**Figure 4.** Ctnna1 is required for proper islet functionality and α-cell survival. (**A**) Heatmap showing the top up-regulated (Group 1) and down-regulated (Group 2) endocrine cell lineage and functional genes in *Ctnna1^{ΔEndo/ΔEndo}* islets from 8-week-old mice. (**B–D**) The figure shows the results of the intraperitoneal glucose tolerance test (**B, D**) and stimulated insulin secretion assay (**C**) performed on control and *Ctnna1^{ΔEndo/ΔEndo}* mice at 8 weeks of age (**B, C**) and 12 months of age (**D**). The *Ctnna1^{ΔEndo/ΔEndo}* mice exhibited glucose intolerance, but no detectable difference in insulin secretion compared to control littermates. The *Ctnna1^{ΔEndo/ΔEndo}* mice are glucose intolerant with non-detectable insulin secretion defects. (**E–F**) Immunofluorescence staining for Gcg, Ins, and DAPI on 8-week-old (**E**) control and (**F**) *Ctnna1^{ΔEndo/ΔEndo}* pancreas sections showing a reduction of α-cells. (**G–H**) Quantification of (**G**) Ins⁺ and (**H**) Gcg⁺ area relative to total pancreas area in 8-week-old control and *Ctnna1^{ΔEndo/ΔEndo}* pancreas sections. (**I–K**) Quantification of endocrine cell area (**I**), Gcg⁺ (**J**), and Sst⁺ (**K**) relative to total pancreas area in control and *Ctnna1^{ΔEndo/ΔEndo}* pancreas sections at P0 and P7. Note that there was no significant difference observed between the control and *Ctnna1^{ΔEndo/ΔEndo}* pancreas sections at P0 and P7. (**L–N**) Immunofluorescence staining for Gcg, TUNEL, and DAPI on P7 control and *Ctnna1^{ΔEndo/ΔEndo}* pancreas sections. White arrows indicate TUNEL⁺/Gcg⁺ cells. The percentage of TUNEL⁺/Gcg⁺ co-positive cells

*Figure 4 continued on next page*

*Figure 4 continued*

relative to total Gcg$^+$ cell numbers in P7 control and *Ctnna1$^{\Delta Endo/\Delta Endo}$* islets is shown in panel (**N**), and compared to controls, the *Ctnna1$^{\Delta Endo/\Delta Endo}$* islets exhibited significantly more TUNEL$^+$/Gcg$^+$ cells at P7. Gcg, glucagon; Ins, insulin; DAPI, 4',6-diamidino-2-phenylindole; IPGTT, intraperitoneal glucose tolerance test; TUNEL, terminal deoxynucleotidyl transferase dUTP nick end labeling; P0, postnatal day 0; P7, postnatal day 7. Data are shown as mean ± SEM. *p<0.05, **p<0.01, ***p<0.001, ****p<0.0001.

The online version of this article includes the following source data and figure supplement(s) for figure 4:

**Source data 1.** Related to *Figure 4B*.

**Source data 2.** Related to *Figure 4C*.

**Source data 3.** Related to *Figure 4D*.

**Source data 4.** Related to *Figure 4G*.

**Source data 5.** Related to *Figure 4H*.

**Source data 6.** Related to *Figure 4I*.

**Source data 7.** Related to *Figure 4J*.

**Source data 8.** Related to *Figure 4K*.

**Source data 9.** Related to *Figure 4N*.

**Figure supplement 1.** Ctnna1 is required for proper islet functionality and normal islet aggregation.

**Figure supplement 1—source data 1.** Related to *Figure 4—figure supplement 1A*.

**Figure supplement 1—source data 2.** Related to *Figure 4—figure supplement 1B*.

**Figure supplement 1—source data 3.** Related to *Figure 4—figure supplement 1C*.

**Figure supplement 1—source data 4.** Related to *Figure 4—figure supplement 1F*.

islets during adulthood is not due to endocrine cell differentiation defects; instead, a drastic increase in apoptosis explains the loss of α-cells in the *Ctnna1*-deleted islets.

## Differential adhesion in endocrine cells controls the formation of islet architecture

Since the reduction of cell-cell adhesion caused distinctive outcomes in different endocrine cell subtypes, we hypothesized that the endocrine cells in the mantle or core of islets may have different adhesiveness. We re-analyzed publicly available gene expression profiles (*DiGruccio et al., 2016*) to examine the expression of key cell adhesion molecules *E-cadherin*, *N-cadherin*, *α-catenin*, and *β-catenin* between the subtypes of endocrine cells. These adhesion genes were expressed at the highest level in β-cells, while their expression was comparatively lower in endocrine cells located in the mantle of islets (α-cells and δ-cells, *Figure 5—figure supplement 1A*). These observations suggest that the adhesive properties may differ between endocrine cell subtypes. According to the differential adhesion hypothesis (DAH), a population of cells with different adhesive properties will tend to spontaneously sort themselves to maximize adhesive bonding (*Foty and Steinberg, 2005*). Stronger adhesion between β-cells may lead them to bundle into islet cores, while the weaker adhesion between non-β-cells results in their dispersion into islet mantles.

Based on the DAH, if the cell adhesion is altered in different endocrine subpopulations, the stereotypical islet architecture will also be affected. To alter the cell adhesion in different endocrine subpopulations, we crossed *Ins1-Cre* and *Gcg-iCre* lines to *Ctnna1-flox* mice and traced the *Ctnna1*-deleted β- or α-cells with the lineage tracer *Rosa26-mT/mG* (*Ins1-Cre; Ctnna1$^{f/f}$; Rosa26-mT/mG* and *Gcg-iCre; Ctnna1$^{f/f}$; Rosa26-mT/mG*, hereafter *Ctnna1$^{\Delta\beta/\Delta\beta}$* and *Ctnna1$^{\Delta\alpha/\Delta\alpha}$* mice, respectively). The *Rosa26-mT/mG* lineage-tracing allele faithfully reported the *Ins1-Cre* and *Gcg-iCre* cell lineages at P7 (*Figure 5A–H*), allowing examination of the respectively labeled β- and α-cells using mGFP expression. The mGFP$^+$ β-cells of *Ctnna1$^{\Delta\beta/+}$* mice are mostly located on the islet core (*Figure 5A and E*), and the mGFP$^+$ α-cells of *Ctnna1$^{\Delta\alpha/+}$* mice are mostly located in the islet mantle (*Figure 5C and G*). Additionally, we found that β-cells exhibited strong F-actin condensed cellular junctions with well-organized rosette structures (*Figure 5—figure supplement 1B*, yellow arrows). In contrast, α-cells displayed loosely bundled F-actin assemblies without focal-condensed junctional structures found in control mice (*Figure 5—figure supplement 1D*, yellow arrows), supporting the notion that α-cells may have relatively low adhesive properties. Inactivation of Ctnna1 in β-cells and α-cells significantly

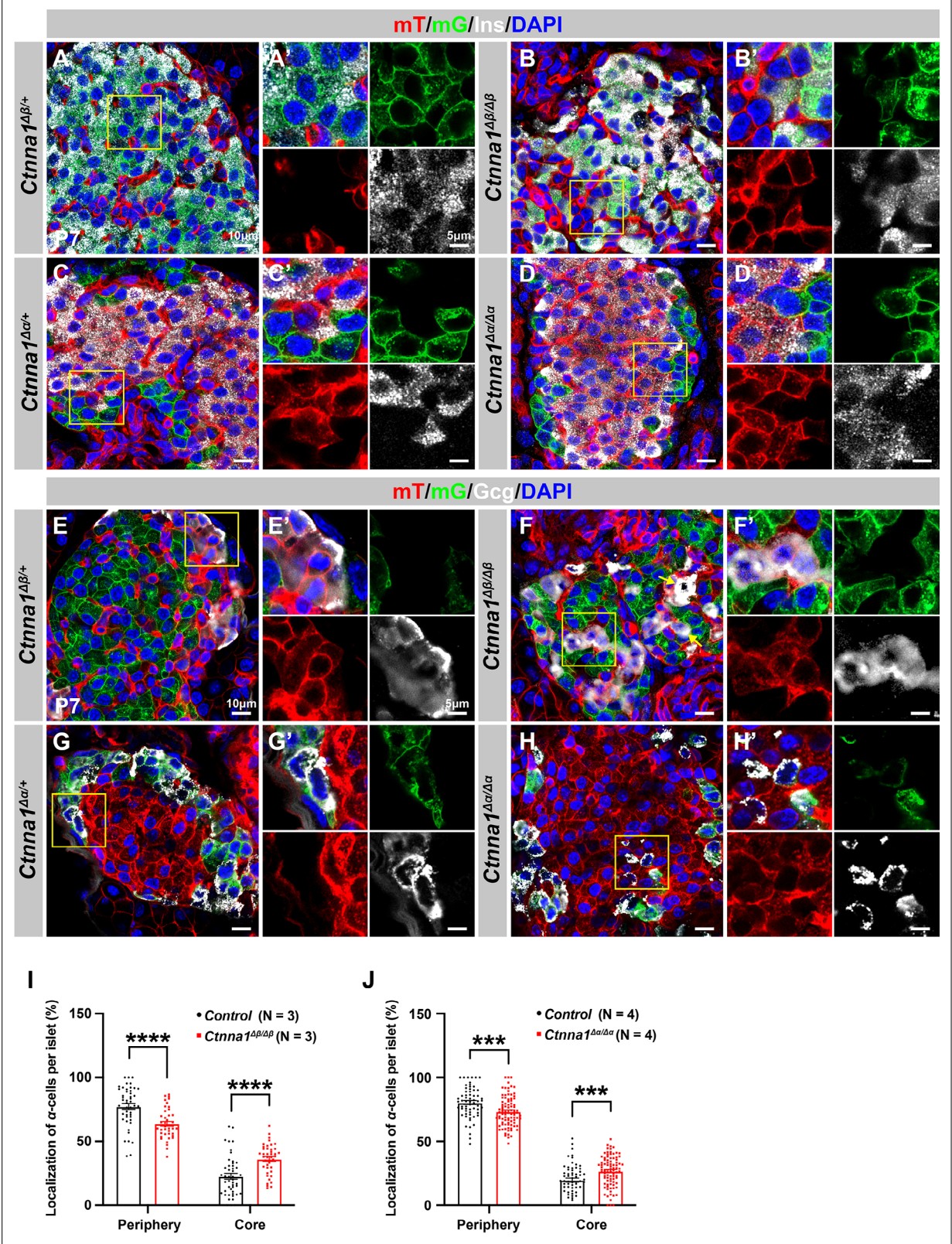

**Figure 5.** Loss of Ctnna1 in α- and β-cells leads to disrupted islet architecture. (**A–H**) Airyscan images of immunofluorescence staining for Ins (**A–D**), Gcg (**E–H**), and DAPI in pancreas sections of P7 mT/mG reporter mice: Heterozygous *Ctnna1^{Δβ/+}* (**A, E**), homozygous *Ctnna1^{Δβ/Δβ}* (**B, F**), heterozygous *Ctnna1^{Δα/+}* (**C, G**), and homozygous *Ctnna1^{Δα/Δα}* (**D, H**). The yellow arrows in (**F**) showing α-cells in the islet core in *Ctnna1^{Δβ/Δβ}* mice. The fields demarcated by yellow boxes are shown at higher magnification with individual color channels in the side panels. (**I–J**) Quantification of α-cell localization

*Figure 5 continued*

in the islets of *Ctnna1^{Δβ/Δβ}* (**I**) and *Ctnna1^{Δα/Δα}* (**J**) mice. Compared to controls, *Ctnna1^{Δβ/Δβ}* and *Ctnna1^{Δα/Δα}* mice exhibit a reduction in peripherally located α-cells and an increase in core-located α-cells. These results suggest that the loss of Ctnna1 in α- and β-cells affects the organization and localization of α-cells within the islet. Gcg, glucagon; DAPI, 4′,6-diamidino-2-phenylindole; Ins, insulin; P7, postnatal day 7. Data are shown as mean ± SEM. ***p<0.001, ****p<0.0001.

The online version of this article includes the following source data and figure supplement(s) for figure 5:

**Source data 1.** Related to *Figure 5I*.

**Source data 2.** Related to *Figure 5J*.

**Figure supplement 1.** Differential cell-cell adhesion in endocrine cells.

diminished the formation of F-actin, as indicated in *Figure 5—figure supplement 1C and E* (highlighted by yellow arrows). This suggests a substantial decrease in cell adhesion assemblies within the β-cells of *Ctnna1^{Δβ/Δβ}* and the α-cells of *Ctnna1^{Δα/Δα}* islets.

Based on the DAH proposed by previous studies, we expected that Ctnna1-expressing α-cells would tend to cluster together and become more concentrated toward the center of *Ctnna1^{Δβ/Δβ}* islets due to weaker adhesion in β-cells lacking Ctnna1. Our observations supported this hypothesis: in *Ctnna1^{Δβ/Δβ}* islets, where only β-cell adhesion was affected but not α-cell adhesion, we observed a significant decrease in α-cells at the periphery and an increase in their density at the core of the islets (*Figure 5B, F and I*). Additionally, our data revealed that islet core-located α-cells had a higher tendency to form clusters (*Figure 5F*). These results are consistent with the prediction of DAH that cells with stronger adhesion tend to exclude those with weaker adhesion and to form a cluster with a defined size.

However, in α-cell-specific Ctnna1 deletion (*Ctnna1^{Δα/Δα}*) mutants, we observed a reduction in α-cells at the periphery and an increase in their frequency in the islet core (*Figure 5D, H and J*), which was unexpected based on the predictions of DAH. Nevertheless, the core-located Ctnna1-deficient α-cells were mostly scattered as single cells without clustering together in the islet of *Ctnna1^{Δα/Δα}* mice (*Figure 5H*). This suggests that the role of Ctnna1 in regulating cellular distribution within the islets may be more complex than the simple attractor-repeller mechanism proposed by DAH. These observations support the idea that differential adhesion between endocrine subtypes is a contributing factor for establishing the islet architecture during development; and the weaker cell adhesion in the peripheral islet layers leads to the specific loss of these cell types in *Ctnna1^{ΔEndo/ΔEndo}* islets.

## Coordination of ECM and cell-cell adhesion regulates islet vascularization, architecture, and functional maturation

Our studies have shown that the coordination of cell-ECM and cell-cell adhesion plays a vital role in regulating islet development. These adhesions work together to balance interactions between endocrine cells and blood vessels. When one of these adhesions is perturbed, islet endocrine cells may compensate with another interaction to maintain islet aggregation (*Figure 3—figure supplement 1*). However, it is still unclear whether islet clustering relies solely on these two adhesions. To test this, we generated *Ngn3-Cre; Ctnna1^{f/f}; Itgb1^{f/f}* (hereafter *Ctnna1; Itgb1^{ΔEndo/ΔEndo}*) double knockout (DKO) mice, in which we simultaneously removed cell-ECM and cell-cell adhesion in all endocrine progenitor cells. We compared control, *Ctnna1^{ΔEndo/ΔEndo}*, *Itgb1^{ΔEndo/ΔEndo}* and DKO islets at P7 to examine their morphology.

At P7, *Ctnna1^{ΔEndo/ΔEndo}* islets displayed loosely aggregated endocrine cells, increased vascularization (*Figure 6A and B*), reduced F-actin assemblies, and more active-Itgb1⁺ endothelial cells in the loosened islet aggregates (*Figure 6—figure supplement 1A and B*). Conversely, *Itgb1^{ΔEndo/ΔEndo}* islets clumped together (*Figure 6C*), with strong F-actin assemblies and less active-Itgb1⁺ endothelial cells in the tightly aggregated islets (*Figure 6—figure supplement 1C*). However, the aggregation of endocrine cells was completely abrogated in the DKO mice (*Figure 6D*). These DKO endocrine cells became suspended single cells scattered throughout the entire pancreatic epithelium, displaying rounded cell morphology (*Figure 6D*). The DKO endocrine cells were larger and displayed weak F-actin assemblies (*Figure 6—figure supplement 1D*). Despite the drastic changes in endocrine cell aggregation and morphology, the endocrine cell subtypes were still present in the P7 DKO islets

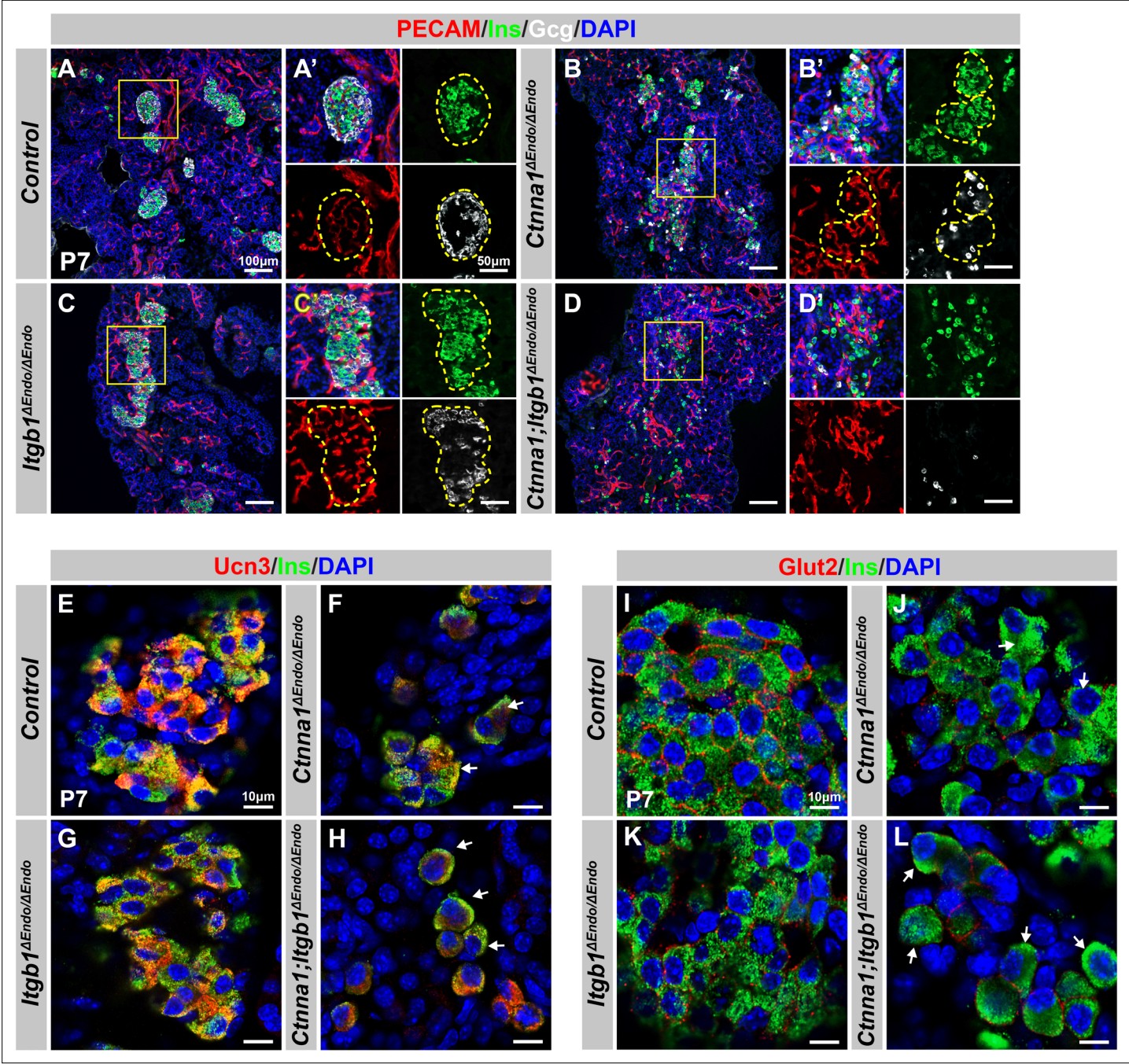

**Figure 6.** Disruption of cell-extracellular matrix (ECM) and/or cell-cell adhesion results in abnormal islet vascular architecture, endocrine cell aggregation, and decreased expression of β-cell maturation markers at postnatal day 7. (**A–D**) Immunofluorescence staining of PECAM, Ins, Gcg, and DAPI in P7 (**A**) control, (**B**) *Ctnna1^ΔEndo/ΔEndo^*, (**C**) *Itgb1^ΔEndo/ΔEndo^*, and (**D**) *Ctnna1; Itgb1^ΔEndo/ΔEndo^* mice. Fields demarcated by yellow boxes in (A–D) are shown at higher magnification in (A'–D') side panels. Individual islet shape is outlined by dashed yellow lines in (A'–C'). Endocrine cells are suspended and scattered throughout the pancreas of *Ctnna1; Itgb1^ΔEndo/ΔEndo^* mice. (**E–L**) Airyscan images of immunofluorescence staining for Ucn3, Glut2, Ins, and DAPI in P7 (**E, I**) control, (**F, J**) *Ctnna1^ΔEndo/ΔEndo^*, (**G, K**) *Itgb1^ΔEndo/ΔEndo^*, and (**H, L**) *Ctnna1; Itgb1^ΔEndo/ΔEndo^* mice. Arrows indicate reduced expression of Ucn3 or Glut2 in β-cells of the mutant mice. PECAM, platelet endothelial cell adhesion molecule-1; Ins, insulin; Gcg, glucagon; DAPI, 4′,6-diamidino-2-phenylindole; Ucn3, urocortin 3; Glut2, Slc2a2; P7, postnatal day 7.

The online version of this article includes the following figure supplement(s) for figure 6:

**Figure supplement 1.** Aberrant endocrine cell clustering and up-regulation of islet progenitor markers upon disruption of cell adhesion.

(*Figure 6D* and data not shown), suggesting that the differentiation of endocrine subtypes is independent of the islet aggregation process.

We next investigated whether the failure in islet aggregation affected endocrine cell maturation by performing IF for β-cell maturation markers. In P7 control mice, Ucn3 and Glut2 were robustly expressed in the cytosol and cell surface of β-cells, respectively (*Figure 6E and I*). Deletion of *Ctnna1* or *Itgb1* in endocrine progenitor cells led to reduced expression of Ucn3 and Glut2 in β-cells (*Figure 6F, G, J, and K*), which was consistent with gene profiling analysis from 8-week-old mice. Finally, DKO islets displayed a severe loss of β-cell maturation markers in P7 β-cells (*Figure 6H and L*). In addition, while endocrine cells in normal islets do not express progenitor markers Sox9 or Spp1 at P7 (*Figure 6—figure supplement 1E and G*), the endocrine cells in the DKO islets exhibited elevated expression of these markers at P7 (*Figure 6—figure supplement 1F and H*). These findings suggest that β-cell maturation is linked to the cell adhesion-mediated islet aggregation process.

To investigate whether the islet aggregation and β-cell maturation defects in DKO mice at P7 were due to developmental delays, we examined islets during later adulthood stages. At 8 weeks, normal endocrine cells form tightly aggregated islets (*Figure 7A*), but in DKO mice, endocrine cells remain mostly scattered, forming smaller and loosely attached clusters (*Figure 7B*). These smaller clusters of endocrine cells abnormally express progenitor markers Sox9 and Spp1 (*Figure 7B*; and data not shown). While the endocrine cells appear to be located closely to each other in lower resolution images, Airyscan super-resolution images of F-actin IF on endocrine cells in DKO islets showed that the endocrine cells are completely separated from each other without forming discernible junction structures (*Figure 7C and D*, yellow arrows). Furthermore, the endocrine cells also exhibit separation from the vasculatures in the DKO islets (*Figure 7C and D*, red arrows). These observations indicate that both the islet aggregation and the β-cell maturation defects in DKO mice persist into adulthood. Supporting this, low expression of β-cell maturation markers Ucn3 and Glut2 persisted in 8-week-old DKO islets (*Figure 7F–H*). These mice were severely glucose intolerant (*Figure 7I*) and exhibit very low insulin secretion upon glucose stimulation (*Figure 7J*). Together, our findings support a model in which islet morphogenesis, vasculature interaction, endocrine cell aggregation, and the β-cell maturation process depend on cell-ECM and cell-cell adhesion during development (*Figure 7K*).

## Discussion

Our study reveals the combinatorial roles of cell-ECM and cell-cell adhesion in regulating the aggregation of endocrine cells into islets. Mechanistically, islet cell aggregation depends on endocrine cell-cell adhesion, which is negatively regulated by cell-ECM adhesion through vascular interaction. These two adhesion properties are reciprocally regulated: lowering cell-ECM adhesion promotes endocrine cell-cell adhesion; and, conversely, lowering cell-cell adhesion promotes cell-ECM adhesion. Importantly, the aggregation and functional maturation processes of endocrine cells are affected by the loss of either adhesion property and are further affected by the loss of both.

In this study, we aimed to elucidate the role of ECM signaling by specifically deleting the essential ECM receptor Itgb1 during endocrine specification. Our findings demonstrate that while the morphology and function of *Itgb1^{ΔEndo/ΔEndo}* islets were impacted, the total islet area remained unaffected, suggesting that Itgb1 deletion may not result in overt cell death in islets, unlike in other organ systems (*Carlson et al., 2008*; *Speicher et al., 2014*). These observations indicate the involvement of other mediators of cell-ECM interactions in islet maintenance. Although Itgb1 is a key player in cell-ECM interactions, compensatory mechanisms involving other integrins, such as β3, β4, β5, β6; and α3, α6, αV integrins, may be operative in Itgb1-deficient islets. Previous studies have highlighted the critical roles of these integrins in cell-ECM interactions and islet biology (*Cirulli et al., 2000*; *Krishnamurthy et al., 2011*; *Schiesser et al., 2021*; *Yashpal et al., 2005*). It is plausible that these integrins compensate for cell-ECM interactions, ensuring the survival and function of Itgb1-deficient islets. Further investigations are warranted to comprehensively understand the compensatory mechanisms and contributions of other integrins in Itgb1-deficient islets.

Both *Itgb1^{ΔEndo/ΔEndo}* and *Itgb1^{ΔMatureβ/ΔMatureβ}* mice demonstrate reduced vascularization, highlighting the importance of ECM signaling in vascular interactions throughout stages from embryonic endocrine cells to mature β-cells. This aligns with previous research using *MIP-Cre* to create *Itgb1*-knockout mice, which also exhibited decreased islet vascularization (*Win et al., 2020*). The critical role of Itgb1 signaling in organ vascularization seems universal; studies indicate that Itgb1 inactivation in pituitary

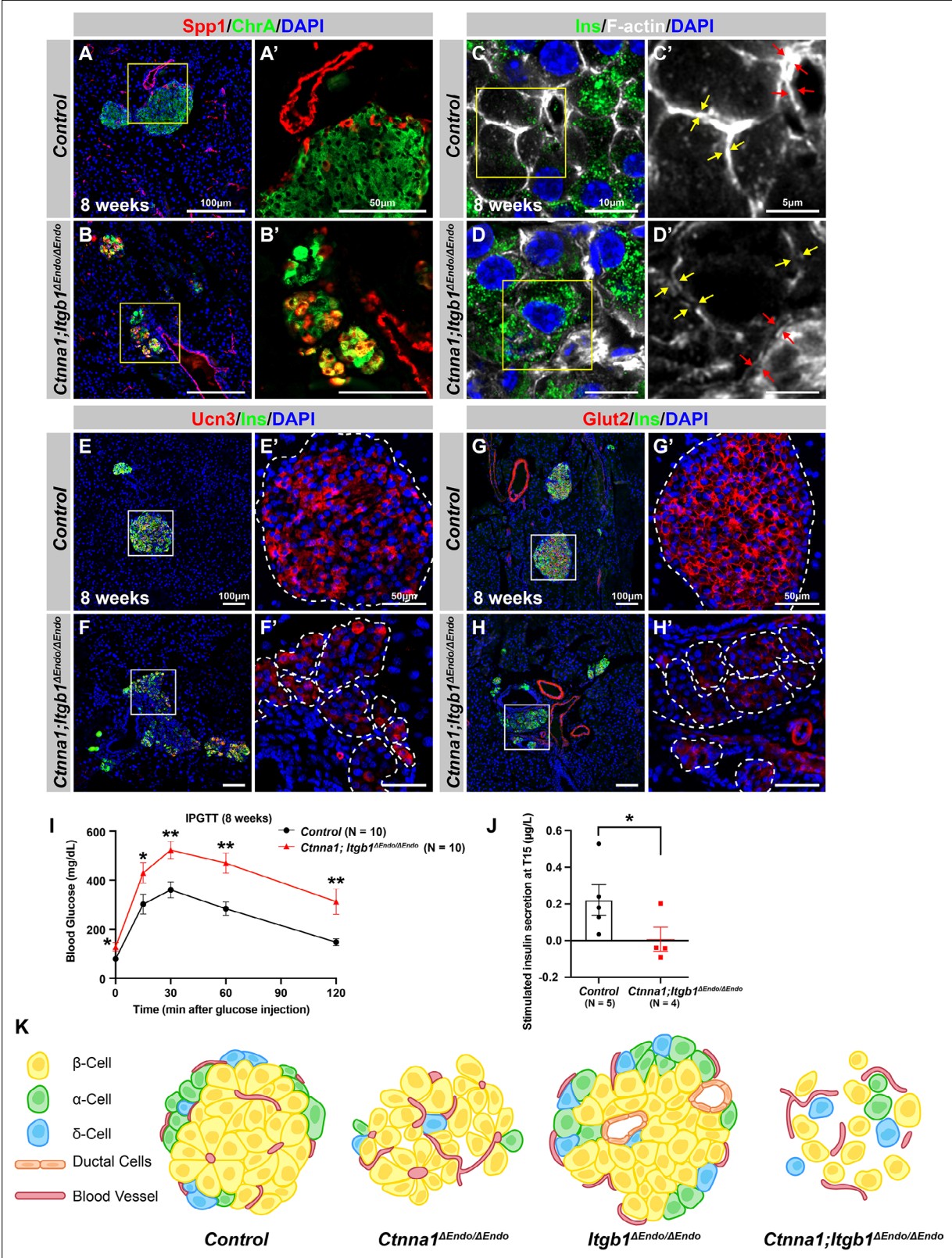

**Figure 7.** Abnormal islet aggregation, mis-regulation of β-cell maturation markers, and insulin secretion defects persist in *Ctnna1; Itgb1^{ΔEndo/ΔEndo}* mice into adulthood. (**A–B**) Immunofluorescence staining for endocrine cell marker ChrA and progenitor marker Spp1 on pancreas sections from control (**A**) and *Ctnna1; Itgb1^{ΔEndo/ΔEndo}* (**B**) mice at 8 weeks of age. The yellow boxes in (A, B) indicate the fields shown at higher magnification in (A', B') side panels. In the islets of *Ctnna1; Itgb1^{ΔEndo/ΔEndo}* mice, a subpopulation of ChrA⁺ endocrine cells expresses progenitor markers Spp1 (yellow cells

*Figure 7 continued on next page*

*Figure 7 continued*

in **B and B'**). (**C–D**) Airyscan images of immunofluorescence staining for Ins, F-actin, and DAPI in pancreatic sections from 8-week-old control and *Ctnna1; Itgb1^{ΔEndo/ΔEndo}* mice are shown. In control islets (**C, C'**), the F-actin assemblies appear aligned with cell membranes and condensed in cellular junctions, as indicated by the yellow and red arrows, respectively. In contrast, diffused distribution of F-actin (red arrows in **D'**) and separation of F-actin assemblies (yellow arrows in **D'**) between Ins⁺ β-cells are observed in the *Ctnna1; Itgb1^{ΔEndo/ΔEndo}* mice. (**E–F**) Immunofluorescence staining of Ucn3, Ins, and DAPI in pancreatic sections of 8-week-old (**E**) control and (**F**) *Ctnna1; Itgb1^{ΔEndo/ΔEndo}* mice. Fields demarcated by white boxes in (E and F) are shown at higher magnification in (**E' and F'**), and individual islet shape is outlined by dashed white lines. The reduction of Ucn3 expression persists in adult *Ctnna1; Itgb1^{ΔEndo/ΔEndo}* mice. (**G–H**) Immunofluorescence staining of Glut2, Ins, and DAPI in 8-week-old (**G**) control and (**H**) *Ctnna1; Itgb1^{ΔEndo/ΔEndo}* mouse pancreas. Fields demarcated by white boxes in (G and H) are shown at higher magnification in (G' and H'), and individual islet shape is outlined by dashed white lines. The reduction of Glut2 expression persists in adult *Ctnna1; Itgb1^{ΔEndo/ΔEndo}* mice. (**I–J**) Intraperitoneal glucose tolerance test was performed on 8-week-old mice (**I**), and glucose-stimulated insulin secretion was measured at T15 in 20- to 38-week-old mice (**J**) from control and *Ctnna1; Itgb1^{ΔEndo/ΔEndo}* groups. *Ctnna1; Itgb1^{ΔEndo/ΔEndo}* mice exhibited severe glucose intolerance and defects in insulin secretion. (**K**) Graphical summary of the islet phenotypes observed after disruption of cell-cell and cell-extracellular matrix (ECM) adhesion. Ins, insulin; ChrA, chromogranin A; Spp1, secreted phosphoprotein 1; Ucn3, urocortin 3; IPGTT, intraperitoneal glucose tolerance test. Data are shown as mean ± SEM. *p<0.05 and **p<0.01.

The online version of this article includes the following source data for figure 7:

**Source data 1.** Related to *Figure 7I*.

**Source data 2.** Related to *Figure 7J*.

endocrine epithelial cells and kidney cells leads to a loss of vasculature (*Mohamed et al., 2020*; *Scully et al., 2016*). Reduced vascular interaction can instigate hypoxia responses, which have been tied to β-cell maturation processes (*Balboa et al., 2022*; *Heinis et al., 2010*; *Zeng et al., 2017*). We thus propose that the impaired β-cell maturation observed in *Itgb1^{ΔEndo/ΔEndo}* mice could stem from a hypoxic state caused by vascular loss. However, this hypothesis requires further empirical investigation for validation.

The expression of *Vegfa*, an essential growth factor for islet vasculature, is not affected in *Itgb1^{ΔEndo/ΔEndo}* islets (*Supplementary file 1*). Thus, the reduced vascularization in Itgb1-deficient islets may not be explained by the lack of Vegfa. Downstream of integrin signaling, integrin-linked kinase (ILK) is activated upon integrin-ECM binding to regulate actin organization, cell migration, and other cellular processes (*Hannigan et al., 2005*). Genetic deletion of ILK in the developing pancreas leads to the failure of ILK-inactivated endocrine cells to adhere to the basement membrane, and a drastic reduction of intra-islet blood vessel density (*Kragl et al., 2016*). This study further showed that cortical actomyosin contraction was significantly increased, thus increasing cortex tension in both the ILK-deleted islets and in Itgb1-inactivated cultured β-cells. Our analyses showed aberrant organization of F-actin fibers and compaction of individual endocrine cells, as well as dysregulated cytoskeleton genes in *Itgb1*-deleted islets. Thus, it is likely that both the *Itgb1^{ΔEndo/ΔEndo}* and *Itgb1^{ΔMatureβ/ΔMatureβ}* islets fail to interact with vasculature resulting in the same alteration of the actomyosin regulation seen in ILK-deleted islet cells. In contrast, the *Ctnna1^{ΔEndo/ΔEndo}* model shows increased islet vascularization without affecting the expression of *Vegfa* (*Supplementary file 2*). We posit that weakened adhesion between endocrine cells in *Ctnna1^{ΔEndo/ΔEndo}* islets may allow greater penetration of vasculature through islet clusters, leading to augmented islet vascularization. Supporting this notion, salivary gland patterning allows vessel penetration via an epithelial 'cleft' created by weakened cell-cell adhesion (*Kwon et al., 2017*). Interestingly, the initial steps of cleft formation requires ECM-Itgb1 signaling (*Sakai et al., 2003*); conversely, reducing cell-cell adhesion promotes cell-ECM adhesion during budding morphogenesis (*Wang et al., 2021*). Thus, our proposed model, in which ECM signaling in concert with cell-cell adhesion controls islet aggregation, may reflect a fundamental mechanism of organ morphogenesis.

In the context of our findings, it is crucial to consider the potential role of vascular alignment in islet cell function. The vasculature system within the islet plays an indispensable role not only in delivering nutrients to the cells but also in facilitating the efficient distribution of hormones into the peripheral circulation (*Gan et al., 2018*). The functional maturation of β-cells and their efficiency in hormone secretion are likely influenced by their proximity to and interaction with the vasculature. In our *Itgb1*- and *Ctnna1*-deficient models, we observed an abnormal aggregation of islet cells, resulting in scattered cells that appear misaligned with the vasculature. This disorganization could potentially impair nutrient delivery and hormone distribution, further exacerbating the functional impairments seen in these models. It is plausible that, in addition to their impact on cell-cell adhesion and aggregation,

the inactivation of Itgb1 and Ctnna1 may also interfere with the proper integration of β-cells with the islet vasculature. This perspective adds another layer to our understanding of the complex interplay between cellular adhesion, aggregation, and the vasculature in islet morphology and function.

The onset of the proendocrine cell program triggers the expression of Ngn3, a factor thought to stimulate the activation of ECM signaling. This signaling pathway is regarded as a key regulatory mechanism controlling the delamination and migration of proendocrine cells away from ductal progenitor cords (*Rosenberg et al., 2010*). In the case of *Itgb1$^{ΔEndo/ΔEndo}$* mice, the deactivation of ECM signaling in these endocrine progenitors leads to a noticeable decrease in their migration distances. Consequently, differentiated endocrine cells remain entangled with ducts post-differentiation from as early as E15.5, highlighting the role of ECM signaling in facilitating endocrine progenitor migration during initial developmental stages. Interestingly, this phenomenon of reduced migration is not observed in *Itgb1$^{ΔMatureβ/ΔMatureβ}$* islets, suggesting the impact of ECM signaling on cell migration may be more specific to the progenitor stage. Supporting this hypothesis, studies utilizing *Ins-Cre* or *RIP-Cre* to ablate Itgb1 in post-differentiated insulin-producing cells have reported no discernible migration defects (*Diaferia et al., 2013*; *Win et al., 2020*). During the limited time window of endocrine cell development, islet cell radial migration is also modulated by semaphorin signaling (*Pauerstein et al., 2017*). The remodeling of actin, induced by semaphorin, is associated with changes in the cell anchorage to the ECM (*Alto and Terman, 2017*). This suggests that the chemoattractant system, where semaphorin signaling occurs, could potentially intersect with the activation of Itgb1 signaling. This intersection of signaling pathways may guide the migration of islet progenitor cells toward the periphery and away from the ducts. Thus, the chemoattractant system where semaphorin signaling occurs may involve the activation of Itgb1 signaling to control islet progenitor cell migration toward the periphery, and away from the ducts. Whether the migration of endocrine progenitors toward the periphery affects their function remains to be tested. However, the role of ECM-integrin signaling extends beyond the scope of cell migration. In adult stages, ECM-integrin signaling continues to have a significant impact on differentiated endocrine cells within pancreatic islets (*Gan et al., 2018*). Therefore, when we use the term 'endocrine cells' in this study, we refer to both progenitor and differentiated cells, and the significance of ECM-integrin signaling spans across multiple cellular processes, not being confined to migration alone.

Despite the endocrine subtype ratios being different in human versus rodent islets, the precise differences in the stereotypical architectures in the two species have been debated. Studies using different imaging modalities have yielded different conclusions (*Dybala et al., 2020*). In mice, it is proposed that spatiotemporal collinearity leads to the typical core-mantle architecture of the spherical islet in which α-cells, the first to develop, form the peninsular outer layer, and β-cells subsequently form beneath them (*Sharon et al., 2019a*). Our current study has not directly tested this hypothesis, yet our observations suggest that differential cell adhesion is critical for developing this structure. Our data support a model in which randomly allocated α- and β-cells are specified and sorted into the final islet architecture based on differential adhesion. It remains to be tested whether the differential cell adhesion is also involved in the development of islet architectures in human. In addition, studies have shown that the Roundabout (Robo)/Slit signaling axis plays a role in determining islet stereotypical architecture. The secreted ligand, Slit, is often associated with the ECM, and binds to Robo receptors to control various cellular responses via cytoplasmic kinases and the regulation of actin and microtubule cytoskeleton (*Jiang et al., 2022*; *Tong et al., 2019*). Mice lacking *Robo1* and *Robo2* in all endocrine cells, or selectively in β-cells, show complete loss of endocrine cell type sorting in the islets (*Adams et al., 2021*; *Adams et al., 2018*; *Gilbert et al., 2020*). Interestingly, Robo1/2 DKO islets exhibit adhesion defects and are prone to dissociate upon isolation (*Adams et al., 2018*), similar to the *Ctnna1$^{ΔEndo/ΔEndo}$* islets. Similar observations have been shown in stem cell sorting and cancer cell migration; binding of Slit to Robo receptors leads to the modulation of Ncad- or Ecad-mediated cell adhesion (*Stine et al., 2014*; *Tong et al., 2019*). We speculate that Robo/Slit signaling regulates endocrine cell type sorting in islets by controlling cell adhesive properties to establish the islet architecture.

## Limitation of the study
Our study's potential limitation is the use of the *Ngn3-Cre* driver for targeted gene deletion. While commonly used for pancreatic endocrine progenitor and its lineage-specific gene deletion, *Ngn3* is

also expressed in other tissues such as the brain and gastrointestinal tract (*Schonhoff et al., 2004*; *Simon-Areces et al., 2010*). Deletion of genes using this driver could impact these other tissues and create unintended consequences. Previous studies show that *Ctnna1* deletion in neuronal progenitor cells in the brain can lead to altered neural cell proliferation and differentiation (*Lien et al., 2006*). Although we did not investigate the potential effects of *Itgb1* and *Ctnna1* deletion in these other tissues, further studies are necessary to comprehend the role of ECM signaling and cell adhesion in these other tissues and the potential consequences of their deletion using the *Ngn3-Cre* driver.

We acknowledge the intricate interplay between cell-ECM and cell-cell interactions and understand that the loss of one type of interaction does not universally enhance the other. This complexity was exemplified in our DKO model, which did not exhibit the expected compensatory behaviors. While our in vitro cell adhesion assay provided some insight, we recognize its limitations in fully mirroring the dynamic and sophisticated changes occurring in vivo. Consequently, our observations serve as preliminary findings that pave the way for more comprehensive investigation. Further research, particularly in more complex, dynamic models, will be vital in elucidating the precise nature of these interactions and their role in cell development and function. The dynamic nature of islet aggregation and endocrine subtype sorting would require more in-depth live imaging analysis. Obtaining 3D cellular information in developing islets would provide insights into these dynamic and complex phenomena, and further our understanding of the process of islet development and morphogenesis. We anticipate that using pancreas slice cultures (*Panzer et al., 2020*) to develop an imaging platform to monitor the development of live neonatal islets would provide a framework for understanding how islets develop, and offer novel ways to harness this understanding in the search for clinical alternatives to treat diabetes.

## Materials and methods

### Key reagents

Details regarding reagents, equipment, and software used in this study are included in the 'Key Resources Table' file.

### Mouse strains

All animal experiments described herein were approved by the City of Hope Institutional Animal Care and Use Committee (Protocol 15041) and Institutional Biosafety Committee (Protocol 16002). Mice carrying *Ngn3-Cre* (*Schonhoff et al., 2004*) (RRID: IMSR_JAX:006333), *Gcg-iCre* (*Shiota et al., 2017*) (RRID:IMSR_JAX:030663), *Itgb1^flox^* (*Raghavan et al., 2000*) (RRID:IMSR_JAX:004605), *Ctnna-1^flox^* (*Vasioukhin et al., 2001*) (RRID:IMSR_JAX:004604), *Ins-Cre* (*Thorens et al., 2015*) (RRID:IMSR_JAX:026801), *Rosa26^mT/mG^* (*Muzumdar et al., 2007*) (RRID:IMSR_JAX007676), *Rosa26^EYFP^* (*Srinivas et al., 2001*) (RRID:IMSR_JAX:006148), and *Ucn3-cre* (*Harris et al., 2014*) (RRID:MMRRC_037417-UCD) alleles have been previously described. Additional information for the mouse lines used in this study is listed in Key Resources Table.

### Whole-mount IF

A standard whole-mount staining protocol was utilized to analyze pancreas 3D morphology, as previously described (*Maldonado et al., 2020*). Briefly, pancreas tissue was dissected and fixed overnight in 4% paraformaldehyde (PFA). The tissue was dehydrated with serial methanol washes, bleached with a Dent's Bleach solution, and rehydrated using an inverse series of methanol washes. Samples were blocked followed by incubation with primary antibodies for PECAM(RRID:AB_393571), ChrA (RRID:AB_789299), and secreted phosphoprotein 1 (Spp1) (RRID:AB_2194992). Samples were then washed in a 0.15% Triton-X-PBS solution and stained with secondary antibodies overnight. Following additional washes with 0.15% Triton-X-PBS, samples were fixed with 4% PFA, and dehydrated with serial ethanol washes. Finally, samples were immersed in a BABB clearing solution (two parts benzyl benzoate and one part benzyl alcohol) and imaged using a Zeiss LSM880 confocal microscope.

### Confocal microscopy

All images were collected with a Zeiss LSM880 confocal microscope equipped with a 10X0.45 NA plan apochromat objective at ×2 magnification, as standard for AiryscanFast collection. All tiled Z-stacks were acquired in AiryscanFast mode, the acquisition method set to Z-stack (per channel), and with

a 2.0 µm step size. Scans were conducted using 405, 488, 561, and 633 nm laser lines, and emissions were collected using standard Airyscan filters and the Airyscan detector, in Zen Black 2.3 SP1 (Carl Zeiss, RRID:SCR_013672). Final voxel sizes were 0.22 × 0.22 × 1.98 µm³, and no digital gain or detector offset was used. Detector gains and laser intensities were consistent across all samples.

## Image segmentation/analysis

Data sets were first Airyscan processed in Zen Black using the 'Auto' setting using a 2D sample image as a base for Batch mode. After processing, the files were converted to IMS format and analyzed in Imaris 9.5 (Bitplane, RRID:SCR_007370). Analysis consisted of manually selecting thresholds that could be consistently applied between samples, and best fit the biology. Due to variations in the depth of the sample, amount of tissue the light had to pass through, and orientation, thresholds were chosen which most accurately represented the objects being analyzed. The ducts were first segmented using a 'Surface' with a 2.00 µm surface grain size, and a fixed threshold. We limited the analysis to areas where ducts were present by segmenting the duct channel a second time (again as a Surface) with a grain size of 10 µm and a much lower threshold, generating a 'bubble' area of interest around the ducts. Within this volume, endocrine positive staining was then segmented as another surface and its distance from the ducts quantified using the median value of the distance transform from the surface of the original ducts. Any endocrine cells contained entirely or mostly within the ducts were considered to have a distance of zero from the ducts.

## Immunofluorescence

Pancreas tissues were collected from littermate control and mutant mice, ensuring consistency in the sample source. IF staining was performed as previously described (*Shih et al., 2007*). Briefly, pancreata were fixed in 4% PFA in PBS, equilibrated in 30% sucrose in PBS, and cryoembedded in Tissue-Tek OCT. To ensure the reliability and accuracy of the findings, we used littermate control and mutant mice, collecting and processing their pancreas samples concurrently. This included fixing and embedding the samples into the same cryo-block, sectioning, and preparing them for IF staining on the same slide. This process ensured identical staining conditions for all samples. Samples were then sectioned at 10 µm intervals from the same cryo-block, facilitating identical staining conditions for all samples. For IF staining, sections were blocked for 1 hr before the addition of primary antibodies diluted in the same buffer. Following washes, secondary antibodies, diluted in blocking buffer, were applied to sections and stained for 2 hr at room temperature. The respective dilutions of primary and secondary antibodies are noted in the Key Resources Table.

Images were taken on a Zeiss Axio-Observer-Z1 microscope with Apotome, and figures were created in Adobe Photoshop/Illustrator CC2020. Morphometric analyses were conducted using Image-Pro Premier v.9.2 (RRID:SCR_016497) and Qupath software v.0.2.3 (RRID:SCR_108257), available at https://qupath.github.io. High-resolution image acquisition was achieved with a Zeiss LSM900 confocal microscope with Airyscan. For images intended for direct comparison, the exposure time was kept constant, and changes in digital gain were avoided to prevent distortion of results. Upon image acquisition, linear adjustment processing was applied uniformly across all comparable samples. These adjustments were made using a linear display with Min/Max rescaling. This adjustment approach was to carefully balance avoiding oversaturation while preserving the accuracy of marker expressions. This approach ensures that the full data range is properly utilized, optimizing the full range of possible intensity values. Specifically, the darkest pixel in the image is adjusted to black (minimum value), the brightest pixel to white (maximum value), and all other pixels are rescaled based on these new extremes.

## α-Cell localization analysis

The islets were imaged from three serial sections per mouse, taken at 150 µm (P7) or 200 µm (8 weeks) intervals. Manual cell counting was performed using QuPath (version 0.2.3). Islets with a minimum of 40 endocrine cells and at least 10 α-cells were included in the analysis. At the postnatal stage, α-cell localization within the islet was determined according to previously described criteria (*Adams et al., 2018*). Peripheral α-cells were defined as those within the first two layers, while α-cells in any layers beyond the first two were considered to be in the core.

## Intraperitoneal glucose tolerance test

Six- to 16-week-old or 6- to 12-month-old mice were fasted for 6–16 hr. Basal blood glucose was measured, and mice were injected intraperitoneally (i.p.) with 2 mg/g body weight dextrose solution. Blood glucose was measured at 15, 30, 60, 90, and 120 min after glucose challenge. The experiments were performed by the Comprehensive Metabolic Phenotyping (CMP) Core at City of Hope.

## Insulin secretion in vivo

The β-cell function was assessed by quantifying insulin secretion in vivo with hyperglycemic clamps. Surgical procedures have been described in detail previously (*Ayala et al., 2011*; *Berglund et al., 2008*). Briefly, mice were anesthetized with isoflurane, and the carotid artery and jugular vein were catheterized. Free catheter ends were tunneled under the skin to the back of the neck and externalized with a MASA (Mouse Antenna for Sampling Access), which permits arterial sampling from an indwelling catheter in conscious mice without handling stress. Mice were individually housed and allowed to recover for 4 days during which body weight returned to within ~10% of presurgical body weight.

For the hyperglycemic clamps, mice were fasted for 5 hr and connected to a swivel. Saline-washed erythrocytes were infused (5–6 µl/min) during the experimental period to prevent a >5% fall in hematocrit. Blood samples were collected from the arterial catheter in tubes containing EDTA and centrifuged, and plasma was stored at −20°C until analyzed. At t=0 min, mice received a priming glucose infusion rate (GIR, 100 mg/kg/min) for 2 min to stimulate first-phase insulin secretion. The GIR was then adjusted to achieve and maintain blood glucose at ~15.0 mmol/l. Blood glucose (5 µl) was measured at t = −15,−5, 5, 10, 15, and 20 min and then every 10 min until t=120 min with an Alpha Trak glucometer (Abbott Laboratories, Abbott Park, IL, USA). Larger blood samples (100 µl) to measure plasma insulin were taken at various timepoints.

## Isolation of mouse pancreatic islets

Pancreatic islets were isolated as previously described (*Taylor et al., 2013*). Briefly, mouse pancreatic islets were isolated using 2.5 mg/ml collagenase (Sigma-Aldrich) as the digestion method. Islets were collected under a stereomicroscope and allowed to recover overnight in PIM(R) 32A (Prodo Labs) medium with 5% HSA, Gentamicin-AmphotericinB, and 5.8 mmol/l glucose.

## RNA-seq

Total RNA was isolated using the RNeasy Micro Kit (QIAGEN) with 10–30 isolated islets and processed per manufacturer's instructions. cDNA libraries were prepared using the KAPA RNA HyperPrep Kit with RiboErase (HMR) according to the manufacturer's instructions. Libraries were subjected to high-throughput sequencing on Illumina HiSeq 2500 (50 bp, single-ended) and subsequent data analysis was completed by the Integrative Genomics Core at City of Hope. For sequence alignment and gene counts, RNA-seq reads were trimmed to remove sequencing adapters using Trimmomatic (*Bolger et al., 2014*) and polyA tails using FASTP (*Chen et al., 2018*). The processed reads were mapped back to the mouse genome (mm10) using STAR software (v. 020201) (*Dobin et al., 2013*). The featureCount software (RRID:SCR_012919) was applied to generate the count matrix, with default parameters (*Liao et al., 2019*). Between two sets of samples (control vs mutant), differential expression analysis was conducted, in R, by adjusting read counts to normalized expression values using TMM normalization method and by correcting batch effects using a design matrix (*Robinson et al., 2010*). Genes with an FDR-adjusted p-value less than 0.05 and with an FC greater than 2 or less than 0.5 were considered as significant up- and down-regulated genes, respectively. Pathway analysis was conducted using DAVID (v. 6.8) (*Huang et al., 2009a*; *Huang et al., 2009b*) and GSEAPreranked algorithm using GSEA Desktop program in Java (*Subramanian et al., 2005*). The former requires a list of up- and down-regulated genes, while the latter a ranked list of whole genes according to their log2 FC and p-values. Raw data and processed data files are available at the GEO database: GSE153187 and GSE190788.

## Quantitative real-time PCR

RNA extraction and purification were performed using the RNeasy Micro Kit (QIAGEN) in accordance with the manufacturer's protocol. The quality and quantity of the total RNA were assessed using a Bioanalyzer 2100 (Agilent Technologies). Samples with an RNA integrity number (RIN) of at least six

were used for cDNA synthesis with the SuperScript First-Strand Synthesis SuperMix for qRT-PCR (Invitrogen). Real-time PCR was conducted using PowerUp SYBR Green Master mix (Applied Biosystems) in a QuantStudio 6 Flex machine (Applied Biosystems). Each reaction contained 1 ng cDNA (total RNA equivalent). Relative mRNA expression levels were calculated by normalizing target genes to *Cyclophilin A* (*Ppia*) as an internal control. All primers were designed using the CDS of the target genes by the PrimerBank online tool (https://pga.mgh.harvard.edu/primerbank/).

## Aggregation assay

Neonatal *Ngn3-Cre; Rosa26mT/mG; Itgb1*<sup>+/f</sup> (control) or *Ngn3-Cre; Rosa26mT/mG; Itgb1*<sup>f/f</sup> (mutant) pancreata were dissociated into a single cell suspension using TrypLE (Thermo Fisher). Enzymatic activity was neutralized using FACS buffer (10% fetal bovine serum and 2 mM EDTA in CMRL media). Cells were passed through a 40 μm cell strainer and DAPI was used to stain dead cells prior to FACS. Prior to cell seeding, 96-well round-bottom plates were coated with either BSA (no ECM) or Matrigel and fibronectin (+ECM). 3000 cells per well were added into no ECM or +ECM wells and cultured for 48 hr.

## Ex vivo pancreas explants

Pancreas explants are described previously (*Shih and Sander, 2014*). After 24 hr of culture, the explants were placed on the microscope stage in 37°C culture chambers with a controlled atmosphere of humidified 5% $CO_2$. Time-lapse imaging was performed using a Zeiss LSM880 inverted confocal microscope with a C-Apochromat 20×/0.75 objective lens. Explants were optically sectioned every 2.5 μm in a 512×512 format with up to 80 μm Z-stacks every 10 min for 72 hr. Images were acquired with Zen software and then reconstructed in 3D with Imaris 9.5 (Bitplane) software. The contrast was adjusted and selected optical planes or *z*-projections of sequential optical sections were used to assemble time-lapse movies.

## Electron microscopy sample preparation

Sample blocks were prepared following the NCMIR protocol (*Deerinck et al., 2010*). Briefly, small pieces (1–2 mm³) of mouse pancreatic tissues were dissected and immediately fixed in 0.2 M cacodylate buffer pH 7.4 containing 2.5% glutaraldehyde and 2% PFA with 3 mM calcium chloride. After primary aldehyde fixation the samples were rinsed in 0.15 M sodium cacodylate pH 7.4 containing 2 mM calcium chloride and post-fixed in 1.5% potassium ferrocyanide-reduced 2% osmium tetroxide in 0.15 M cacodylate buffer for 1 hr. Tissue was then rinsed in distilled water and treated with 0.1% aqueous thiocarbohydrazide for 20 min. After further rinsing in distilled water the tissue was again treated with 2% osmium tetroxide for 30 min, rinsed in distilled water, dehydrated in an ethanol series, and infiltrated with Durcupan ACM resin. Ultra-thin sections (70 nm thick) were acquired by ultramicrotomy and examined on an FEI Tecnai 12 transmission electron microscope equipped with a Gatan OneView CMOS camera.

## Statistics

Statistical analyses were conducted using GraphPad Prism (RRID:SCR_002798). Unless otherwise specified, comparisons between groups were made using a one-way analysis of variance (ANOVA) and a two-tailed Student's t-test. Each condition was tested with at least three independent measurements. Except where otherwise indicated, all data are represented as the mean ± standard error of the mean (SEM).

Every experiment was independently replicated in the laboratory a minimum of three times. Samples were assigned to experimental groups either pairwise (for wild-type cells with or without treatment) or based on their genotypes (wild-type vs knockout). Detailed information about the replicates, including the number of replicates and sample sizes, can be found in the figure legends and is also outlined in *Supplementary file 3*.

Raw data and statistical calculations for intraperitoneal glucose tolerance test, glucose-stimulated insulin secretion, morphometric measurements, and cell population counts for each figure are systematically organized and made available in the 'Source Data' files within the Supplementary files.

## Acknowledgements

We acknowledge the support of City of Hope Core Facilities: Dr. Brain Armstrong from the Microscopy Core, Dr. Zhuo Li from the Electron Microscopy Core, Drs. Xiwei Wu and Min-Hsuan Chen from the Integrative Genomics Core, Dr. Patrick Fueger from the Comprehensive Metabolic Phenotyping Core at City of Hope. We thank Dr. Maike Sander for sharing the anti-Ngn3 antiserum and Dr. Rupangi Vasa-vada for sharing the *Ins-Cre* mouse strain. The California Institute for Regenerative Medicine (CIRM) Training Grant EDUC4-12772 to WT; JDRF postdoctoral fellowship 3-PDF-2019-742-A-N to MM; NSERC Discovery Grant (RGPIN-2016-04276), CIHR New Investigator Award Msh-147794, and MSFHR Scholar Award 18309 to JLK, NIH/NIDDK 1R01DK110276 to MOH; NIH/NIDDK 1R01DK120523, Human Islet Research Network (HIRN) New investigator Award (subaward via UC4DK104162), and a grant from the Wanek Family Foundation to Cure Type 1 Diabetes to SD; Wanek Family Foundation to Cure Type 1 Diabetes and NIH/NIDDK 1R01DK119590 to HPS. We acknowledge the assistance of OpenAI's ChatGPT in refining the language and grammar of our manuscript during resubmission. The contributions of this AI model were strictly limited to textual improvements, with no role in generating or modifying the scientific content and findings of the study.

## Additional information

### Funding

| Funder | Grant reference number | Author |
| --- | --- | --- |
| California Institute for Regenerative Medicine | Training Grant EDUC4-12772 | Wilma Tixi |
| JDRF | Postdoctoral fellowship 3-PDF-2019-742-A-N | Maricela Maldonado |
| Natural Sciences and Engineering Research Council of Canada | Discovery Grant RGPIN-2016-04276 | Janel L Kopp |
| Canadian Institutes of Health Research | New Investigator Award Msh-147794 | Janel L Kopp |
| Michael Smith Foundation for Health Research | Scholar Award 18309 | Janel L Kopp |
| National Institute of Diabetes and Digestive and Kidney Diseases | 1R01DK110276 | Mark O Huising |
| National Institute of Diabetes and Digestive and Kidney Diseases | 1R01DK120523 | Sangeeta Dhawan |
| Human Islet Research Network | New investigator award UC4DK104162 | Sangeeta Dhawan |
| Wanek Family Foundation to Cure Type 1 Diabetes | | Sangeeta Dhawan Hung Ping Shih |
| National Institute of Diabetes and Digestive and Kidney Diseases | 1R01DK119590 | Hung Ping Shih |

The funders had no role in study design, data collection and interpretation, or the decision to submit the work for publication.

### Author contributions

Wilma Tixi, Maricela Maldonado, Data curation, Formal analysis, Investigation, Writing - review and editing; Ya-Ting Chang, Data curation, Formal analysis, Investigation; Amy Chiu, Formal analysis, Visualization, Writing - review and editing; Wilson Yeung, Data curation, Formal analysis, Writing - review and editing; Nazia Parveen, Michael S Nelson, Formal analysis; Ryan Hart, Patrick Fueger, Data curation, Formal analysis; Shihao Wang, Wu Jih Hsu, Data curation; Janel L Kopp, Data curation, Funding acquisition, Writing - review and editing; Mark O Huising, Sangeeta Dhawan, Formal analysis, Funding

acquisition, Writing - review and editing; Hung Ping Shih, Conceptualization, Data curation, Supervision, Funding acquisition, Investigation, Writing - original draft, Project administration, Writing - review and editing

### Author ORCIDs
Maricela Maldonado (ID) http://orcid.org/0000-0003-4682-6900
Michael S Nelson (ID) http://orcid.org/0000-0003-0480-5597
Wu Jih Hsu (ID) http://orcid.org/0000-0001-5385-9079
Janel L Kopp (ID) http://orcid.org/0000-0002-1875-3401
Hung Ping Shih (ID) http://orcid.org/0000-0003-3035-5841

### Ethics

All animal experiments described herein were approved by the City of Hope Institutional Animal Care and Use Committee (Protocol 15041) and Institutional Biosafety Committee (Protocol 16002).

### Decision letter and Author response

Decision letter https://doi.org/10.7554/eLife.90006.sa1
Author response https://doi.org/10.7554/eLife.90006.sa2

---

# Additional files

### Supplementary files

• Supplementary file 1. List of Itgb1-regulated genes in 6-week-old islets. This table provides a comprehensive catalogue of gene expression differences observed between islets isolated from control mice and those from $Itgb1^{\Delta Endo/\Delta endo}$ mice.

• Supplementary file 2. List of Ctnna1-regulated genes in 8-week-old islets. This table provides a comprehensive catalogue of gene expression differences observed between islets isolated from control mice and those from $Ctnna1^{\Delta Endo/\Delta Endo}$ mice.

• Supplementary file 3. List of replicates and statistical analysis. This table provides detailed information about the replicates, including the number of replicates and sample sizes. It also includes the types of statistical analyses performed.

• MDAR checklist

### Data availability

Sequencing data have been deposited in GEO under accession numbers GSE153187 (https://www.ncbi.nlm.nih.gov/geo/query/acc.cgi?acc=GSE153187) and GSE190788 (https://www.ncbi.nlm.nih.gov/geo/query/acc.cgi?acc=GSE190788). In addition, two spreedsheets of the RNA-seq data are provided as *Supplementary files 1 and 2*. Source data files have been provided for all figures. Each experiment was replicated in the laboratory at least three times. Samples were allocated into experimental groups either pairwise (for WT cells with or without treatment) or based on the genotypes (WT vs KO). All the replicate information and the number of replicates and sample sizes can be found in figure legends or in *Supplementary file 3*.

The following datasets were generated:

| Author(s) | Year | Dataset title | Dataset URL | Database and Identifier |
|---|---|---|---|---|
| Shih H, Chang Y | 2023 | The role of Itgb1 in islet development | https://www.ncbi.nlm.nih.gov/geo/query/acc.cgi?acc=GSE153187 | NCBI Gene Expression Omnibus, GSE153187 |
| Tixi W, Shih H | 2022 | Investigating the role of cell adhesion molecule alpha-catenin during islet development | https://www.ncbi.nlm.nih.gov/geo/query/acc.cgi?acc=GSE190788 | NCBI Gene Expression Omnibus, GSE190788 |

The following previously published dataset was used:

| Author(s) | Year | Dataset title | Dataset URL | Database and Identifier |
|---|---|---|---|---|
| DiGruccio MR, Mawla AM, Donaldson CJ, Noguchi GM, Vaughan J, Cowing-Zitron C, van der Meulen T, Huising MO | 2016 | Comprehensive alpha, beta and delta cell transcriptomes reveal that ghrelin selectively activates delta cells and promotes somatostatin release from pancreatic islets | https://www.ncbi.nlm.nih.gov/geo/query/acc.cgi?acc=GSE80673 | NCBI Gene Expression Omnibus, GSE80673 |

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

# Appendix 1

## Appendix 1—key resources table

| Reagent type (species) or resource | Designation | Source or reference | Identifiers | Additional information |
|---|---|---|---|---|
| Antibody | Goat polyclonal anti-Pancreatic Polypeptide/PP | Novus Biologicals | Cat# NB100-1793 RRID: AB_2268669 | IHC (1:1000) |
| Antibody | Mouse monoclonal anti-Glucagon | Sigma-Aldrich | Cat# G2654 RRID: AB_259852 | IHC (1:500) |
| Antibody | Goat anti-Glucagon Antibody (N-17) | Santa Cruz Biotechnology | Cat# sc-7780 RRID: AB_641025 | IHC (1:500) |
| Antibody | Guinea Pig polyclonal anti-Insulin | Dakocytomation | Cat# A0564 RRID: AB_10013624 | IHC (1:2000) |
| Antibody | Rabbit monoclonal anti-Insulin | Cell Signaling Technology | Cat# 3014 S RRID: AB_2126503 | IHC & WMIHC (1:1000) |
| Antibody | Rabbit polyclonal anti-Chromogranin A | Novus Biologicals | Cat# NB120-15160 RRID: AB_789299 | IHC & WMIHC (1:500) |
| Antibody | Goat polyclonal anti-Chromogranin A | Santa Cruz Biotechnology | Cat# sc-1488 RRID: AB_2276319 | IHC & WMIHC (1:500) |
| Antibody | Goat polyclonal anti-Somatostatin | Santa Cruz Biotechnology | Cat# sc-7819 RRID: AB_2302603 | IHC (1:100) |
| Antibody | Rat monoclonal anti-Mouse CD31 | BD Biosciences | Cat# 550274 RRID: AB_393571 | IHC & WMIHC (1:200) |
| Antibody | Goat polyclonal anti-PECAM-1 | R&D Systems | Cat# AF3628 RRID: AB_2161028 | IHC (1:500) |
| Antibody | Goat polyclonal anti-Osteopontin/OPN (Spp1) | R&D Systems | Cat# AF808 RRID: AB_2194992 | IHC & WMIHC (1:1000) |
| Antibody | Guinea Pig polyclonal anti-Neurogenin3 | A generous gift of Maike Sander (*Henseleit et al., 2005*) | | IHC (1:1000) |
| Antibody | Mouse monoclonal anti-Nkx6.1 | DSHB | Cat# F55A10 RRID: AB_532378 | IHC (1:500) |
| Antibody | Rabbit polyclonal anti-Pdx1 | Abcam | Cat# ab47267 RRID: AB_777179 | IHC (1:500) |
| Antibody | Rabbit polyclonal anti-Laminin (Lam1) | Sigma-Aldrich | Cat# L9393 RRID: AB_477163 | IHC (1:1000) |
| Antibody | Rat monoclonal anti-EPCAM | DSHB | Cat# G8.8 RRID: AB_2098655 | IHC (1:100) |
| Antibody | Goat polyclonal anti-E-cadherin | R&D Systems | Cat# AF748 RRID: AB_355568 | IHC (1:500) |
| Antibody | Rabbit polyclonal anti-MafA | Abcam | Cat# ab26405 RRID: AB_776146 | IHC (1:200) |
| Antibody | Rabbit polyclonal anti-Urocortin3 (Ucn3) | A generous gift of Mark Huising (*van der Meulen et al., 2015*) | | IHC (1:1000) |
| Antibody | Goat polyclonal anti-Glut2 | Santa Cruz Biotechnology | Cat# sc-7580 RRID: AB_641066 | IHC (1:500) |
| Antibody | Rat monoclonal anti-CD29 (Itgβ1) | BD Biosciences | Cat# 550531 RRID: AB_393729 | IHC (1:500) |
| Antibody | Rabbit monoclonal anti-Ki67 | Cell Signaling | Cat# 12202 S (D3B5) RRID: AB_2620142 | IHC (1:500) |
| Antibody | Rabbit anti-Sox9 antibody | EMD Millipore | Cat# AB5535 RRID: AB_2239761 | IHC (1:500) |
| Antibody | Rabbit polyclonal anti-ERO1b antiserum | A generous gift of David Ron (*Zito et al., 2010*) | | IHC (1:300) |
| Antibody | Cy3 AffiniPure Donkey Anti-Goat IgG (H+L) (donkey polyclonal) | Jackson ImmunoResearch | Cat# 705-165-147 RRID: AB_2307351 | IHC (1:2000) |
| Antibody | Alexa Fluor 488 AffiniPure Donkey Anti-Goat IgG (H+L) (donkey polyclonal) | Jackson ImmunoResearch | Cat# 705-545-147 RRID: AB_2336933 | IHC (1:1000) |

*Appendix 1 Continued on next page*

*Appendix 1 Continued*

| Reagent type (species) or resource | Designation | Source or reference | Identifiers | Additional information |
|---|---|---|---|---|
| Antibody | Alexa Fluor 647 AffiniPure Donkey Anti-Goat IgG (H+L) (donkey polyclonal) | Jackson ImmunoResearch | Cat# 705-605-147 RRID: AB_2340437 | IHC (1:200–500) |
| Antibody | Cy3 AffiniPure Donkey Anti-Guinea Pig IgG (H+L) (donkey polyclonal) | Jackson ImmunoResearch | Cat# 706-165-148 RRID: AB_2340460 | IHC (1:2000) |
| Antibody | Alexa Fluor 488 AffiniPure Donkey Anti-Guinea Pig IgG (H+L) (donkey polyclonal) | Jackson ImmunoResearch | Cat# 706-545-148 RRID: AB_2340472 | IHC (1:1000) |
| Antibody | Alexa Fluor 647 AffiniPure Donkey Anti-Guinea Pig IgG (H+L) (donkey polyclonal) | Jackson ImmunoResearch | Cat# 706-605-148 RRID: AB_2340476 | IHC (1:200–500) |
| Antibody | Cy3 AffiniPure Donkey Anti-Rabbit IgG (H+L) (donkey polyclonal) | Jackson ImmunoResearch | Cat# 711-165-152 RRID: AB_2307443 | IHC (1:2000) |
| Antibody | Alexa Fluor 488 AffiniPure Donkey Anti-Rabbit IgG (H+L) (donkey polyclonal) | Jackson ImmunoResearch | Cat# 711-545-152 RRID: AB_2313584 | IHC (1:1000) |
| Antibody | Alexa Fluor 647 AffiniPure Donkey Anti-Rabbit IgG (H+L) (donkey polyclonal) | Jackson ImmunoResearch | Cat# 711-605-152 RRID: AB_2492288 | IHC (1:200–500) |
| Antibody | Cy3 AffiniPure Donkey Anti-Rat IgG (H+L) (donkey polyclonal) | Jackson ImmunoResearch | Cat# 712-165-153 RRID: AB_2340667 | IHC (1:2000) |
| Antibody | Alexa Fluor 488 AffiniPure Donkey Anti-Rat IgG (H+L) (donkey polyclonal) | Jackson ImmunoResearch | Cat# 712-545-153 RRID: AB_2340684 | IHC (1:1000) |
| Antibody | Alexa Fluor 647 AffiniPure Donkey Anti-Rat IgG (H+L) (donkey polyclonal) | Jackson ImmunoResearch | Cat# 712-605-153 RRID: AB_2340694 | IHC (1:200–500) |
| Commercial assay, kit | Dolichos Biflorus Agglutinin (DBA), Biotinylated | Vector Laboratories | Cat# B1035 RRID: AB_2314288 | IHC (1:500) |
| Commercial assay, kit | Dolichos Biflorus Agglutinin (DBA), Rhodamine | Vector Laboratories | Cat# RL-1032–2 RRID: AB_2336396 | IHC (1:500) |
| Commercial assay, kit | DAPI (4',6-Diamidino-2-Phenylindole, Dihydrochloride) | ThermoFisher Scientific | Cat# D1306 | IHC (1:2000) |
| Commercial assay, kit | Phalloidin-iFluor 647 Reagent | Abcam | Cat# ab176759 | IHC (1:1000) |
| Commercial assay, kit | Alexa Fluor 647 Streptavidin | Jackson ImmunoResearch | Cat# 016-600-084 | IHC (1:500) |
| Commercial assay, kit | Cy3 Streptavidin (for IHC 1:500) | Jackson ImmunoResearch | Cat# 016-160-084 RRID: AB_2337244 | IHC (1:500) |
| Chemical compound, drug | Tween 20, Fisher BioReagents | Fisher Scientific | BP337-500 | |
| Chemical compound, drug | Triton X-100 | Sigma | X100-1L | |
| Chemical compound, drug | VectaShield Mounting Medium for Fluorescence | Vector Laboratories | H-1000 | |
| Chemical compound, drug | Paraformaldehyde | Sigma-Aldrich | P6148-500G | |
| Chemical compound, drug | Methanol | VWR Chemical | BDH1135-4LP | |
| Chemical compound, drug | DMSO | ATCC | 4 X-5 | |
| Chemical compound, drug | Hydrogen Peroxide, ACS, 30%, Stabilized | VWR Chemical | BDH7690-1 | |
| Chemical compound, drug | Benzyl Benzoate (ACROS organics) | Fisher Scientific | AC105862500 | |
| Chemical compound, drug | Benzyl Alcohol | Fisher Scientific | 100-51-6 | |
| Chemical compound, drug | D-Glucose | Mallinckrodt | 4192 | |
| Chemical compound, drug | Bovine Albumin Fraction V (7.5% Solution) | Thermo Fisher | 15260037 | |
| Chemical compound, drug | TrypLE Express, with phenol red, Gibco | Fisher Scientific | 12605–010 | |

*Appendix 1 Continued on next page*

*Appendix 1 Continued*

| Reagent type (species) or resource | Designation | Source or reference | Identifiers | Additional information |
|---|---|---|---|---|
| Chemical compound, drug | EDTA, 0.5 M, pH 8.0 | Corning | 46–034 CI | |
| Chemical compound, drug | Fetal Bovine Serum | Biofluid Technologies | BT-101–500-D | |
| Chemical compound, drug | Heparin sodium salt from porcine intestinal mucosa | Sigma-Aldrich | H3149-100KU | |
| Chemical compound, drug | CMRL 1066, Supplemented CIT Modification | Mediatech | 98–304-CV | |
| Chemical compound, drug | RNase-Free DNase Set | Qiagen | 79254 | |
| Chemical compound, drug | Penicillin-Streptomycin | Gibco | 15070–063 | |
| Chemical compound, drug | DMEM/F12, HEPES, no phenol red | Gibco | 11-039-21 | |
| Chemical compound, drug | Roche Blocking | Roche | 17091700 | |
| Chemical compound, drug | Donor Donkey Serum | Gemini Bio | 100-151-500 | |
| Chemical compound, drug | Cacodylic Acid, Sodium Salt, trihydrate | TedPella Inc | 18851 | |
| Chemical compound, drug | Glutaraldehyde, 25% EM grade | TedPella Inc | 18426 | |
| Chemical compound, drug | Thiocarbohydrazide ≥98.0% | VWR International | TCT1136-25G | |
| Chemical compound, drug | Buffer RLT Plus Lysis Buffer | Qiagen | 1053393 | |
| Chemical compound, drug | 2-Mercaptoethanol | Sigma-Aldrich | M3148-25ML | |
| Chemical compound, drug | Ethanol, Absolute 200 proof | Fisher BioReagents | BP2818100 | |
| Peptide, recombinant protein | Fibronectin Human Protein, Plasma | Gibco | 33016–015 | |
| Commercial assay, kit | AlphaTrak2 Blood Glucose Test Strips | Zoetis | 71681–01 | |
| Commercial assay, kit | AlphaTrak2 Meter | Zoetis | 71676–01 | |
| Commercial assay, kit | Streptavidin/Biotin Blocking Kit | Vector Laboratories | SP-2002 | |
| Commercial assay, kit | Rneasy Micro Kit | Qiagen | 74304 | |
| Commercial assay, kit | Rnase Zap | Invitrogen | AM9780 | |
| Commercial assay, kit | KAPA RNA HyperPrep Kit with RiboErase (HMR) | Roche | KK8560 | |
| Commercial assay, kit | AxyPrep Mag PCR Clean-up kit | Axigen | MAG-PCR-CL-1 | |
| Commercial assay, kit | NovaSeq 6000 Reagent Kit | Illumina | | |
| Commercial assay, kit | DeadEnd Fluorometric TUNEL System | Promega | G3250 | |
| Commercial assay, kit | M.O.M. (Mouse on Mouse) Immunodetection Kit, Basic | Vector Laboratories | BMK-2202 | |
| Commercial assay, kit | PowerUp SYBR Green Master mix | Applied Biosystems | A25742 | |
| Commercial assay, kit | SuperScript First-Strand Synthesis SuperMix for qRT-PCR | Invitrogen | 11752–050 | |
| Commercial assay, kit | Ultrasensitive Mouse Insulin ELISA | Mercodia | 10-1249-01 | |
| Other | RNA sequencing data of Itgb1 KO murine pancreatic islets. | https://www.ncbi.nlm.nih.gov/geo/ | GSE153187 | |
| Other | RNA sequencing data of Ctnna1 KO murine pancreatic islets. | https://www.ncbi.nlm.nih.gov/geo/ | GSE190788 | |

*Appendix 1 Continued on next page*

*Appendix 1 Continued*

| Reagent type (species) or resource | Designation | Source or reference | Identifiers | Additional information |
|---|---|---|---|---|
| Strain, strain background (*M. musculus*) | Mouse: B6.FVB(Cg)-Tg(Neurog3-cre)C1Able/J | Jackson Laboratory, Bar Harbor, ME | RRID:IMSR_JAX:006333 | |
| Strain, strain background (*M. musculus*) | Mouse: B6;129-Itgb1<sup>tm1Efu</sup>/J | Jackson Laboratory, Bar Harbor, ME | RRID:IMSR_JAX:004605 | |
| Strain, strain background (*M. musculus*) | Mouse: B6;129-Ctnna1<sup>tm1Efu</sup>/J | Jackson Laboratory, Bar Harbor, ME | RRID:IMSR_JAX:004604 | |
| Strain, strain background (*M. musculus*) | Mouse: B6;129S-Gcg<sup>tm1.1(icre)Gkg</sup>/J | Jackson Laboratory, Bar Harbor, ME | RRID:IMSR_JAX:030663 | |
| Strain, strain background (*M. musculus*) | Mouse: B6(Cg)-Ins1<sup>tm1.1(cre)Thor</sup>/J | Jackson Laboratory, Bar Harbor, ME | RRID:IMSR_JAX:026801 | |
| Strain, strain background (*M. musculus*) | Mouse: B6.129(Cg)-Gt(ROSA)26Sor<sup>tm4(ACTB-tdTomato,-EGFP)Luo</sup>/J | Jackson Laboratory, Bar Harbor, ME | RRID:IMSR_JAX:007676 | |
| Strain, strain background (*M. musculus*) | Mouse: B6.129X1-Gt(ROSA)26Sor<sup>tm1(EYFP)Cos</sup>/J | Jackson Laboratory, Bar Harbor, ME | RRID:IMSR_JAX:006148 | |
| Strain, strain background (*M. musculus*) | B6.FVB(Cg)-Tg(Ucn3-cre) KF43Gsat/Mmucd | Mutant Mouse Resource & Research Centers, Novi, MI | RRID:MMRRC_037417-UCD | |
| Sequence-based reagent | Generic Cre | The Jackson Laboratory | PCR Primer oIMR1084 | GCG TCG TGG CAG TAA AAA CTA TC |
| Sequence-based reagent | Generic Cre | The Jackson Laboratory | PCR Primer oIMR1085 | GTG AAA CAG CAT TGC TGT CAC TT |
| Sequence-based reagent | Generic Cre | The Jackson Laboratory | PCR Primer oIMR7338 | CTA GGC CAC AGA ATT GAA AGA TCT |
| Sequence-based reagent | Generic Cre | The Jackson Laboratory | PCR Primer oIMR7339 | GTA GGT GGA AAT TCT AGC ATC ATC C |
| Sequence-based reagent | β1 integrin floxed | The Jackson Laboratory | PCR Primer oIMR1906 | CGG CTC AAA GCA GAG TGT CAG TC |
| Sequence-based reagent | β1 integrin floxed | The Jackson Laboratory | PCR Primer oIMR1907 | CCA CAA CTT TCC CAG TTA GCT CTC |
| Sequence-based reagent | α-catenin floxed | The Jackson Laboratory | PCR Primer oIMR1902 | CAT TTC TGT CAC CCC CAA AGA CAC |
| Sequence-based reagent | α-catenin floxed | The Jackson Laboratory | PCR Primer oIMR1903 | GCA AAA TGA TCC AGC GTC CTG GG |
| Sequence-based reagent | Gcg<sup>iCre</sup> | The Jackson Laboratory | PCR Primer oIMR7338 | CTA GGC CAC AGA ATT GAA AGA TCT |
| Sequence-based reagent | Gcg<sup>iCre</sup> | The Jackson Laboratory | PCR Primer oIMR7339 | GTA GGT GGA AAT TCT AGC ATC ATC C |
| Sequence-based reagent | Gcg<sup>iCre</sup> | The Jackson Laboratory | PCR Primer oIMR9266 | AGA TGC CAG GAC ATC AGG AAC CTG |
| Sequence-based reagent | Gcg<sup>iCre</sup> | The Jackson Laboratory | PCR Primer oIMR9267 | ATC AGC CAC ACC AGA CAC AGA GAT C |
| Sequence-based reagent | Rosa<sup>mT/mG</sup> | The Jackson Laboratory | PCR Primer 9655 | CCA GGC GGG CCA TTT ACC GTA AG |
| Sequence-based reagent | Rosa<sup>mT/mG</sup> | The Jackson Laboratory | PCR Primer oIMR8545 | AAA GTC GCT CTG AGT TGT TAT |
| Sequence-based reagent | Rosa<sup>mT/mG</sup> | The Jackson Laboratory | PCR Primer oIMR8546 | GGA GCG GGA GAA ATG GAT ATG |
| Sequence-based reagent | R26R-EYFP | The Jackson Laboratory | PCR Primer 21306 | CTG GCT TCT GAG GAC CG |
| Sequence-based reagent | R26R-EYFP | The Jackson Laboratory | PCR Primer 24500 | CAG GAC AAC GCC CAC ACA |
| Sequence-based reagent | R26R-EYFP | The Jackson Laboratory | PCR Primer 24951 | AGG GCG AGG AGC TGT TCA |
| Sequence-based reagent | R26R-EYFP | The Jackson Laboratory | PCR Primer 24952 | TGA AGT CGA TGC CCT TCA G |
| Sequence-based reagent | mItgb1-F | PrimerBank | qPCR primer Primer Bank ID: 52,722 a1 | ATGCCAAATCTTGCGGAGAAT |

*Appendix 1 Continued on next page*

*Appendix 1 Continued*

| Reagent type (species) or resource | Designation | Source or reference | Identifiers | Additional information |
|---|---|---|---|---|
| Sequence-based reagent | mItgb1-R | PrimerBank | qPCR primer Primer Bank ID: 52,722 a1 | TTTGCTGCGATTGGTGACATT |
| Sequence-based reagent | mSlc2a2-F | PrimerBank | qPCR primer Primer Bank ID: 13654262 a1 | TCAGAAGACAAGATCACCGGA |
| Sequence-based reagent | mSlc2a2-R | PrimerBank | qPCR primer Primer Bank ID: 13654262 a1t | GCTGGTGTGACTGTAAGTGGG |
| Sequence-based reagent | mUcn3-F | PrimerBank | qPCR primer Primer Bank ID: 21492632 a1 | AAGCCTCTCCCACAAGTTCTA |
| Sequence-based reagent | mUcn3-R | PrimerBank | qPCR primer Primer Bank ID: 21492632 a1 | GAGGTGCGTTTGGTTGTCATC |
| Sequence-based reagent | mMafA-F | PrimerBank | qPCR primer Primer Bank ID: 23503735 a1 | AGGAGGAGGTCATCCGACTG |
| Sequence-based reagent | mMafA-R | PrimerBank | qPCR primer Primer Bank ID: 23503735 a1 | CTTCTCGCTCTCCAGAATGTG |
| Sequence-based reagent | mMafB-F | PrimerBank | qPCR primer Primer Bank ID: 23308601 a1 | TTCGACCTTCTCAAGTTCGACG |
| Sequence-based reagent | mMafB-R | PrimerBank | qPCR primer Primer Bank ID: 23308601 a1 | TCGAGATGGGTCTTCGGTTCA |
| Sequence-based reagent | mArx-F | PrimerBank | qPCR primer Primer Bank ID: 26024213 a1 | GGCCGGAGTGCAAGAGTAAAT |
| Sequence-based reagent | mArx-R | PrimerBank | qPCR primer Primer Bank ID: 26024213 a1 | TGCATGGCTTTTTCCTGGTCA |
| Sequence-based reagent | mEtv1-F | PrimerBank | qPCR primer Primer Bank ID: 26328055 a1 | TTAAGTGCAGGCGTCTTCTTC |
| Sequence-based reagent | mEtv1-R | PrimerBank | qPCR primer Primer Bank ID: 26328055 a1 | GGAGGCCATGAAAAGCCAAA |
| Sequence-based reagent | mAldh1a3-F | PrimerBank | qPCR primer Primer Bank ID: 31542123 a1 | GGGTCACACTGGAGCTAGGA |
| Sequence-based reagent | mAldh1a3-R | PrimerBank | qPCR primer Primer Bank ID: 31542123 a1 | CTGGCCTCTTCTTGGCGAA |
| Sequence-based reagent | mHk1-F | PrimerBank | qPCR primer Primer Bank ID: 309289 a1 | CGGAATGGGGAGCCTTTGG |
| Sequence-based reagent | mHk1-R | PrimerBank | qPCR primer Primer Bank ID: 309289 a1 | GCCTTCCTTATCCGTTTCAATGG |
| Sequence-based reagent | mLdhA-F | PrimerBank | qPCR primer Primer Bank ID: 6754524 a1 | TGTCTCCAGCAAAGACTACTGT |
| Sequence-based reagent | mLdhA-R | PrimerBank | qPCR primer Primer Bank ID: 6754524 a1 | GACTGTACTTGACAATGTTGGGA |
| Sequence-based reagent | mCtnna1-F | PrimerBank | qPCR primer Primer Bank ID: 6753294 a1 | AAGTCTGGAGATTAGGACTCTGG |
| Sequence-based reagent | mCtnna1-R | PrimerBank | qPCR primer Primer Bank ID: 6753294 a1 | ACGGCCTCTCTTTTTATTAGACG |
| Sequence-based reagent | mSlc27a2-F | PrimerBank | qPCR primer Primer Bank ID: 6755548 a1 | TCCTCCAAGATGTGCGGTACT |
| Sequence-based reagent | mSlc27a2-R | PrimerBank | qPCR primer Primer Bank ID: 6755548 a1 | TAGGTGAGCGTCTCGTCTCG |
| Sequence-based reagent | mPeg10-F | PrimerBank | qPCR primer Primer Bank ID: 31376257 a1 | TGCTTGCACAGAGCTACAGTC |
| Sequence-based reagent | mPeg10-R | PrimerBank | qPCR primer Primer Bank ID: 31376257 a1 | AGTTTGGGATAGGGGCTGCT |
| Sequence-based reagent | mSst-F | PrimerBank | qPCR primer Primer Bank ID: 6678035 a1 | ACCGGGAAACAGGAACTGG |
| Sequence-based reagent | mSst-R | PrimerBank | qPCR primer Primer Bank ID: 6678035 a1 | TTGCTGGGTTCGAGTTGGC |
| Sequence-based reagent | mAldh1a1-F | PrimerBank | qPCR primer Primer Bank ID: 7304881 a1 | ATACTTGTCGGATTTAGGAGGCT |

*Appendix 1 Continued on next page*

*Appendix 1 Continued*

| Reagent type (species) or resource | Designation | Source or reference | Identifiers | Additional information |
|---|---|---|---|---|
| Sequence-based reagent | mAldh1a1-R | PrimerBank | qPCR primer Primer Bank ID: 7304881 a1 | GGGCCTATCTTCCAAATGAACA |
| Sequence-based reagent | mCxcl14-F | PrimerBank | qPCR primer Primer Bank ID: 9625004 a1 | GAAGATGGTTATCGTCACCACC |
| Sequence-based reagent | mCxcl14-R | PrimerBank | qPCR primer Primer Bank ID: 9625004 a1 | CGTTCCAGGCATTGTACCACT |
| Sequence-based reagent | mPPIA-Fw | PrimerBank | qPCR primer Primer Bank ID: 6679438 c1 | GAGCTGTTTGCAGACAAAGTTC |
| Sequence-based reagent | mPPIA-Rv | PrimerBank | qPCR primer Primer Bank ID: 6679438 c1 | CCCTGGCACATGAATCCTGG |
| Software, algorithms | Image-Pro Premier v.9.2 | https://www.bioimager.com/product/image-pro-premier-software/ | RRID:SCR_016497 | |
| Software, algorithms | Imaris Microscopy Image Analysis Software | https://imaris.oxinst.com/ | RRID:SCR_007370 | |
| Software, algorithms | QuPath Quantitative Pathology & Bioimage Analysis | https://qupath.github.io/ | RRID:SCR_018257 | |
| Software, algorithms | ZEISS ZEN Microscope Software | http://www.zeiss.com/microscopy/en_us/products/microscope-software/zen.html#introduction | RRID:SCR_013672 | |
| Software, algorithms | Star software (v.020201) | http://code.google.com/p/rna-star/ | RRID:SCR_004463 | |
| Software, algorithms | DAVID software (v.6.8) | http://david.abcc.ncifcrf.gov/ | RRID:SCR_001881 | |
| Software, algorithms | Featurecount software | http://bioinf.wehi.edu.au/featureCounts/ | RRID:SCR_012919 | |
| Software, algorithms | R-Project for Statistical Computing | http://www.r-project.org/ | RRID:SCR_001905 | |
| Software, algorithms | GraphPad Prism Statistical Analysis Software | http://www.graphpad.com/ | RRID:SCR_002798 | |
| Software, algorithms | Applied Biosystems QuantStudio 6 Real Time PCR System - QuantStudio Real-Time PCR Software v1.3 | https://www.thermofisher.com/us/en/home/global/forms/life-science/quantstudio-6-7-flex-software.html | RRID: SCR_020239 | |

