## [Editor Report]

This important study advances our knowledge regarding how islet endocrine cells interact with one another and with surrounding blood vessels during embryonic development and in adult mice. The evidence supporting this work is convincing, with complementary microscopy, functional and transcriptomic data. The authors' conclusions are supported by the data with sample size and quantitative analyses. These data should be of broad interest to islet biologists.

---

## [Decision Letter]

**Decision letter after peer review:**

[Editors’ note: the authors submitted for reconsideration following the decision after peer review. What follows is the decision letter after the first round of review.]

Thank you for submitting the paper "Coordination between ECM and cell-cell adhesion regulates the development of islet aggregation, architecture, and functional maturation" for consideration by *eLife*. Your article has been reviewed by 3 peer reviewers, and the evaluation has been overseen by a Reviewing Editor and a Senior Editor. The following individuals involved in the review of your submission has agreed to reveal their identity: Rebecca Hull-Meichle (Reviewer #2); Vijay Yechoor (Reviewer #3).

Comments to the Authors:

We are sorry to say that, after consultation with the reviewers, we have decided that this work will not be considered further for publication by *eLife*.

Overall, the reviewers thought the study was of interest and had the potential to provide novel insights into how cell-matrix and cell-cell interactions contribute to the development and function of pancreatic islets. However, there were several issues raised about experimental numbers, statistical analysis, and image quality. Additional experiments would also be needed to support the conclusions related to the role of hypoxia and cell-ECM interactions. There is also a need for a more extensive analysis of the RNA-Seq data and rationale for following up on CTNNA1. More extensive analysis is especially important given some of the discrepancies of this study's findings/interpretations with past studies in the literature.

*Reviewer #1 (Recommendations for the authors):*

The authors of this manuscript conducted a series of sophisticated gene deletion experiments that aimed to understand the relative contribution of cell-cell and cell-matrix interactions in the development and function of pancreatic islets. Despite the importance of the questions being asked in their study, the data are over-interpreted, and some conclusions are not supported by the data presented or by the current literature on the subject. Specific examples include the misinterpretation of basic mechanisms of cell interactions such as the model of differential adhesion suggesting that Ctnna1-null β-cells merge into the core of islet clusters due to weaker adhesion. The model proposed by Foty and Steinberg (validated in numerous developing epithelia) predicts the opposite, i.e. cells with weaker adhesion are pushed outward by cells with stronger adhesion.

In experiments reporting on the phenotype of double KO animals in which both α-catenin and beta1 integrin were simultaneously deleted, the interpretation of the results is clearly affected by the profound alterations of cell-cell and cell-matrix interactions, making it difficult to draw conclusions about normal pancreas development.

Strengths include the sophisticated animal models permitting cell lineage tracing experiments for live imaging, and state-of-the-art techniques used for the analysis of the pancreatic phenotype. However, the manuscript presents data that are either incompletely analyzed or insufficient to support the authors' claims. As a result, it is unlikely that this study, in its current presentation, will significantly advance our knowledge of the function of these two distinct mechanisms of cell interactions in pancreas development.

The authors of this manuscript conducted a series of sophisticated gene deletion experiments that aimed to understand the relative contribution of cell-cell and cell-matrix interactions in the development and function of pancreatic islets.

Despite the importance of the questions being asked in their study, as presented, the manuscript presents several weaknesses.

Specific examples are listed below:

The cell lineage tracing experiments with the mT/mG reporter are superficially described, and the data presented are not vetted by solid statistical analysis.

The authors do not provide any information about the number of mice studied for each animal model, the number of tissue sections stained and analyzed, number of experiments performed. This is critical information that should be provided both in the methods and in the legend of the figures.

Most immunofluorescence images provided in the figures are oversaturated, making it difficult to discriminate, at least qualitatively, levels of expression of the different markers.

In Figure 2, data shown in panel N are difficult to interpret given the large scale on the Y axis, for the relatively low counts reported in the plot. Is it really possible to have a *** statistical significance between control cells and integrin beta1 KO when the values are almost identical? What type of statistical method was used? It is also unclear how many of such experiments were performed (n=?).

In the manuscript, (lines 433-435), the authors state: "… ECM-Integrin signaling is critical for endocrine cell migration, and that the time window for endocrine cell initiation is small." According to the data presented, and some of the literature cited, it is the "endocrine progenitors" and not the "endocrine cells" that rely on the ECM-Integrin signaling for cell migration. It is also inaccurate to state that ECM-Integrin signaling does not occur in endocrine cells once they have differentiated. Thus, while the function of this interaction may or may not translate into cell migration, integrin signaling continues to play important roles in differentiated cells in multiple tissues.

Experiments conducted with the Ucn3-Cre;Itgb1f/f mice are superficially described. A more detailed presentation and discussion of the phenotype of these mice is necessary to draw conclusions about the role of Itgb1 in β cells after they have differentiated. This is especially important for comparison with other studies if one considers that could reveal important information compared to previous studies that used the insulin promoter as a driver of Cre. This is important since the Ucn3 and Insulin genes are expressed at different times and at different levels during β cell development.

The RNA-seq data, although interesting, are only superficially analyzed, nor validated. Experiments of immunofluorescence do not really validate the most interesting alterations of genes found affected by the gene deletions, since it is not very quantitative. It would have been helpful to confirm at the protein level by Western blotting the expression of some of the most interesting genes, or at least by qPCR. The authors would be surprised to discover that RNA-seq does not necessarily translate into the same order of magnitude of mRNA levels when testing the same samples by qPCR.

*Reviewer #2 (Recommendations for the authors):*

The goal of this study was to uncover mechanisms that govern the aggregation and "self-assembly" of islets during embryonic development and postnatally, a process that is not well understood. The authors focused on the role of integrin-β 1 (Itgb1), a key mediator of cell-extracellular matrix (ECM) interactions, and on α-catenin (Ctnna1), a hub protein that links cell adhesion molecules with the cytoskeleton. They used conditional knockout models (Ngn3-Cre and Ucn3-Cre), microscopy and morphometric image analysis, and transcriptomic, and functional measures of glucose metabolism. They found that loss of Itgb1 from endocrine precursor cells resulted in premature aggregation of islets and impaired migration of islets from pancreatic ducts resulting in disturbed islet morphology. Additionally, they observed loss of islet vascularization and decreased expression of markers of endocrine cell identity, which together resulted in glucose intolerance. On the other hand, loss of Ctnna1 resulted in smaller, hypervascularized islets which also expressed fewer markers of differentiated endocrine cell identity resulting in an age-dependent loss of glucose tolerance. Ctnna1 mice also exhibited loss of glucagon and somatostatin-positive islets cells, with loss of glucagon+ cells occurring postnatally. The authors concluded from these studies that a balance exists between cell-ECM and cell-cell interactions, and that loss of one of these interactions (as occurs in Itgb1 and Ctnna1 deficient mice, respectively) may result in an increase in the other. However, the loss of both Itgb1 and Ctnna1 from endocrine precursor cells resulted in an almost complete loss of islet aggregation and severe glucose intolerance.

The conclusions of this paper are mostly supported by the data. Strengths include the thorough characterization of mice, especially the Itgb1 mice with the Ngn3-Cre driver. The work shows a clear effect of both Itgb1 and Ctnna1 to modulate islet aggregation/morphology without affecting cell specification or overall abundance (during development). The study provides compelling evidence for the role of these cell adhesion molecules in governing islet aggregation, morphology, identity, and function and is impactful for the field.

The weaknesses are generally minor, with support for some statements being made using a small amount of data (single images in some cases); although most of these do not detract from the main message of the work. For the in vitro aggregation study in the Itgb1 deficient cells, while true that these cells seem to be able to form small aggregates after short-term plating on ECM, further aggregation was inhibited. This illustrates that other mediators of cell-ECM interactions may also be important, a point that was not fully acknowledged in this study. The work on cell shape and cell-cell adhesion in the Itgb1 mice is not as strongly supported by data as the other components of that work. For the Ctnna1 work, the details regarding F-actin localization are hard to appreciate from the images provided. The concept that glucagon+ cells may undergo cell death due to detachment from surrounding cells is exciting, but it is not clear whether the current data fully support that hypothesis.

Together, these data show that both cell-ECM and cell-cell interactions are critical for normal aggregation, morphology, and vascularization of islets and that disturbances in either, or both, have substantial effects on cell identity and function. This work will be impactful for the islet biology field.

*Reviewer #3 (Recommendations for the authors):*

Pancreatic islet morphogenesis is a complex interplay between endocrine cell-intrinsic signaling coupled with external cues from ECM and neighboring cells. Defining the specific pathways and the developmental window when they are critical to the eventual islet architecture and function still remains unclear. This study adds to the body of evidence that the ECM-Integrin signaling and cell-cell adhesion signaling mediated via α-catenin has a critical combinatorial role in determining the islet cell aggregation, vascularization, and eventual function.

Strengths:

The reciprocal regulation of the ECM-integrin pathway and cell-cell adhesion via the α-catenin pathway identified in this study provides insight into how the external cues for developing endocrine cells are processed and integrated to eventually allow mouse islets to develop a core of β-cells and the mantle comprising of other endocrine cells. The detailed dissection to identify the developmental window using in vivo studies complemented with explant studies with lineage tracing is compelling strength of this study.

The predicted changes in mouse islet (mantle-core) architecture due to changes in cell-cell adhesion strength as validated by the combination of the models that the authors use in this study provides a better understanding of the dynamics of islet morphogenesis.

Weakness:

The authors suggest that hypoxia resulting from changes in the vasculature is a potential cause of functional immaturity stemming from the expression of disallowed genes and enrichment of genes in RNA-seq involved in Hypoxic responses. Experimental demonstration of hypoxia and its causative role in these models is however lacking to substantiate this explanation.

Many other groups have previously demonstrated the importance of α-catenin and β-integrin 1 in islet morphogenesis and function, using earlier (Pdx1 promoter-driven Cre) and later (various insulin promoter-driven Cre) developmental windows. Though this study narrows down the window to when the endocrine lineage is determined by Ngn3 and suggests that the altered cell adhesion leads to changes in function and aggregation, intracellular signaling pathways suggested by other studies, such as the sonic hedgehog pathway, PARP, and others remain unexplored in the current context to provide more mechanistic insight.

While the regulation of the vascularization of the islets in both these mutant models (Itfb1 and Ctnna1) individually and in combination is very convincing, the authors note that this was with no change in angiogenic factor expression. While their explanation invoking the alterations of cortical tension by changes in actin cytoskeleton is certainly plausible and intriguing, it is not demonstrated experimentally and remains to be established in future work.

1. Concurrent measurement of insulin and glucose during the GTT provides compelling evidence of a β-cell defect. Adding this with the GTT data in Figures 1, 4, and S6 would add to the strength of the study.

2. The glucose and functional phenotype of the Ucn-Cre mediated Itgb1ΔMatureβ/ΔMatureβ has not been described. Since this is a new model generated by the authors, it is important to describe the phenotype and also compare and contrast it to the other previously described mature β-cell ITGB knockouts that the authors allude to in their discussion.

3. Page 6, Line 219 – Should be 'Figure 2K, 2L' and not 'Figure 1K, 1l.'

---

## [Author Response]

[Editors’ note: the authors resubmitted a revised version of the paper for consideration. What follows is the authors’ response to the first round of review.]

Reviewer #1 (Recommendations for the authors):The authors of this manuscript conducted a series of sophisticated gene deletion experiments that aimed to understand the relative contribution of cell-cell and cell-matrix interactions in the development and function of pancreatic islets. Despite the importance of the questions being asked in their study, the data are over-interpreted, and some conclusions are not supported by the data presented or by the current literature on the subject. Specific examples include the misinterpretation of basic mechanisms of cell interactions such as the model of differential adhesion suggesting that Ctnna1-null β-cells merge into the core of islet clusters due to weaker adhesion. The model proposed by Foty and Steinberg (validated in numerous developing epithelia) predicts the opposite, i.e. cells with weaker adhesion are pushed outward by cells with stronger adhesion.

We would like to express our gratitude to the reviewer for taking the time to carefully review our manuscript and for providing valuable feedback. We apologize for any confusion that may have arisen from our previous statements and for any misinterpretation that may have occurred. To clarify, we had never stated that the model of differential adhesion suggests that *Ctnna1*-null β-cells merge into the core of islet clusters due to weaker adhesion.

As we stated in our previous submission, our findings support the differential adhesion hypothesis (DAH), which predicts that weaker adhesion in non-β-cells will lead to their dispersion into the islet mantles, while stronger adhesion between β-cells may lead to them bundling into islet cores. We believe that our findings were largely aligned with the differential adhesion hypothesis. Our prediction that Ctnna1-expressing α-cells would cluster together and become more concentrated towards the center of *Ctnna1^Δβ/Δβ^* islets due to weaker adhesion in β-cells lacking Ctnna1 was based on the principle that cells with different adhesive properties will tend to sort themselves to maximize adhesive bonding. We have now provided supporting evidence to validate this prediction, including a statistical analysis that clearly shows a significant decrease in α-cells at the periphery and an increase in their density at the core of the islets (Figure 5B, 5F, 5I).

We also quantified the cellular location in α-cell-specific Ctnna1-deletion (*Ctnna1^Δα/Δα^*) islets and observed a reduction in α-cells at the periphery and an increase in their frequency in the islet core (Figure 5D, 5H, 5J). While this was unexpected based on the predictions of DAH, we acknowledge that the role of Ctnna1 in regulating cellular distribution within the islets may be more complex than the simple attractor-repeller mechanism proposed by DAH. Our observation that the core-located Ctnna1-deficient α-cells were mostly scattered as single cells without clustering together in the islet of *Ctnna1^Δα/Δα^* mice (Figure 5H) supports this idea.

Overall, our findings support the concept that differential adhesion between endocrine subtypes is a contributing factor in establishing the islet architecture during development. The weaker cell adhesion in the peripheral islet layers leads to the specific loss of these cell types in *Ctnna1*^Δ*Endo/*Δ*Endo*^ islets. We appreciate the opportunity to address the reviewer's concerns and have revised our manuscript accordingly to provide more clarity and evidence to support our conclusions.

In experiments reporting on the phenotype of double KO animals in which both α-catenin and beta1 integrin were simultaneously deleted, the interpretation of the results is clearly affected by the profound alterations of cell-cell and cell-matrix interactions, making it difficult to draw conclusions about normal pancreas development.Strengths include the sophisticated animal models permitting cell lineage tracing experiments for live imaging, and state-of-the-art techniques used for the analysis of the pancreatic phenotype. However, the manuscript presents data that are either incompletely analyzed or insufficient to support the authors' claims. As a result, it is unlikely that this study, in its current presentation, will significantly advance our knowledge of the function of these two distinct mechanisms of cell interactions in pancreas development.

We thank the reviewer for his/her insightful comment. We greatly appreciate the depth of reviewer’s analysis concerning the complex cell-cell and cell-matrix interactions in our *Itgb1; Ctnna1 DKO* mice model, and fully acknowledge the potential challenges these factors could introduce when interpreting our results. It appears, however, that there might be a small misapprehension regarding the specific focus of our research. Our study is primarily oriented towards understanding islet development, utilizing the *Ngn3-Cre* driver expressed in pancreatic endocrine progenitors. We did not intend, in our prior submission, to make broader conclusions about the overall development of the pancreas. We absolutely concur with the reviewer's intriguing suggestion about the interrelation of islet morphogenesis and the formation of the exocrine pancreas. Indeed, we share this curiosity, and it forms the basis of our follow-up study. However, as this question extends beyond the boundaries of the present report, we chose not to include these aspects in the current manuscript.

We sincerely hope this provides a clearer understanding of our study's intent and specificity. If there are any further points the reviewer would like us to elaborate on, we would be more than pleased to do so.

The authors of this manuscript conducted a series of sophisticated gene deletion experiments that aimed to understand the relative contribution of cell-cell and cell-matrix interactions in the development and function of pancreatic islets.Despite the importance of the questions being asked in their study, as presented, the manuscript presents several weaknesses.Specific examples are listed below:The cell lineage tracing experiments with the mT/mG reporter are superficially described, and the data presented are not vetted by solid statistical analysis.The authors do not provide any information about the number of mice studied for each animal model, the number of tissue sections stained and analyzed, number of experiments performed. This is critical information that should be provided both in the methods and in the legend of the figures.

In our initial submission, we included the Materials Design Analysis Reporting (MDAR) Checklist and provided a Source Data spreadsheet in an Excel format. In these two supplementary documents, the details of the statistical analyses and all raw data used for each figure were provided. However, we understand that these details may have been overlooked in the main text/legends etc., and we acknowledge reviewer’s concerns.

To address these concerns, we have made several revisions. Firstly, we have provided all relevant details, including statistical numbers and individual data points, in the figure labels or legends. Additionally, we have revised the “List of Replicates and Statistical Analysis” and “Source Data spreadsheet “ in Supplementary files to assist reviewers in accessing the raw data points. We hope that these revisions will provide a clearer understanding of our methods and results and address reviewer’s concerns. We appreciate the feedback provided by the reviewer, which has helped us improve the quality of our manuscript.

Most immunofluorescence images provided in the figures are oversaturated, making it difficult to discriminate, at least qualitatively, levels of expression of the different markers.

We appreciate the reviewer's insightful observation concerning the oversaturation of the immunofluorescence images in our initial submission. We understand that this might have complicated the qualitative discrimination of different marker expression levels.

In response to this valuable feedback, we now present several new images and have also performed readjustments to the intensity of most existing ones. These adjustments were made using a linear display with Min/Max rescaling, a common image processing technique that adjusts the intensity values in the images. Our aim with this adjustment approach was to carefully balance avoiding oversaturation while preserving the accuracy of marker expressions. This approach ensures that the full data range is properly utilized, optimizing the full range of possible intensity values. Specifically, the darkest pixel in the image is adjusted to black (minimum value), the brightest pixel to white (maximum value), and all other pixels are rescaled based on these new extremes. This approach enhances the contrast and visibility of features in the images, making them more suitable for publication or presentation. We believe that these modifications significantly improve the visual clarity of our results, facilitating a more straightforward qualitative comparison of different marker expressions.

We would also like to emphasize the meticulous semi-quantitative approach employed in our study. To ensure the reliability and accuracy of our findings, we used littermate control and mutant mice, collecting and processing their pancreas samples concurrently. This included fixing and embedding the samples into the same cryo-block, sectioning, and preparing them for immunofluorescence staining on the same slide. This process ensured identical staining conditions for all samples.

During the imaging process, we took particular care to maintain consistency. For images intended for direct comparison, we kept the exposure time constant and avoided the use of any digital gain that could potentially distort our results. Upon image acquisition, we applied linear adjustment processing uniformly across all comparable samples. This rigorous methodology provides a robust foundation for our comparative analysis, enabling us to confidently attribute observed differences to genetic variations rather than procedural inconsistencies. We also provide this information in our revised method section under ‘Immunofluorescence (IF)’ paragraph.

In Figure 2, data shown in panel N are difficult to interpret given the large scale on the Y axis, for the relatively low counts reported in the plot. Is it really possible to have a *** statistical significance between control cells and integrin beta1 KO when the values are almost identical? What type of statistical method was used? It is also unclear how many of such experiments were performed (n=?).

We appreciate the reviewer's keen observation regarding Figure 2N from our previous submission. We agree that the Y-axis scale may have presented difficulties in data interpretation. For clarity, we have revised this figure, using the same data set in the resubmission. This data was obtained from an equal number of control and mutant mice (three each). The statistical analysis was performed utilizing GraphPad Prism software. Group comparisons were made using one-way analysis of variance (ANOVA) and a two-tailed student's t-test. To enhance transparency, all figures have been revised to include biological replicates (N). All raw data, replicates and numbers are provided in the "List of Replicates and Statistical Analysis" and “Source Data spreadsheet” in Supplementary files. We are confident that these revisions will offer clearer insight into our methodology and results.

In the manuscript, (lines 433-435), the authors state: "… ECM-Integrin signaling is critical for endocrine cell migration, and that the time window for endocrine cell initiation is small." According to the data presented, and some of the literature cited, it is the "endocrine progenitors" and not the "endocrine cells" that rely on the ECM-Integrin signaling for cell migration. It is also inaccurate to state that ECM-Integrin signaling does not occur in endocrine cells once they have differentiated. Thus, while the function of this interaction may or may not translate into cell migration, integrin signaling continues to play important roles in differentiated cells in multiple tissues.

We appreciate the reviewer's detailed and insightful comments. We realize that our language in the cited lines could lead to confusion, and we regret any inaccuracies in our statements. Indeed, it is the "endocrine progenitors," rather than the "endocrine cells," that heavily rely on ECM-Integrin signaling for their migration. We also acknowledge that ECM-Integrin signaling continues to be active in differentiated endocrine cells, and it plays crucial roles in islets as many other studies have shown.

In the revised manuscript, we have corrected these inaccuracies and provided a more accurate reflection of the roles of ECM-Integrin signaling in both endocrine progenitors and differentiated endocrine cells. We appreciate the reviewer's attention to detail in ensuring the precision and clarity of our work.

Experiments conducted with the Ucn3-Cre;Itgb1f/f mice are superficially described. A more detailed presentation and discussion of the phenotype of these mice is necessary to draw conclusions about the role of Itgb1 in β cells after they have differentiated. This is especially important for comparison with other studies if one considers that could reveal important information compared to previous studies that used the insulin promoter as a driver of Cre. This is important since the Ucn3 and Insulin genes are expressed at different times and at different levels during β cell development.

We appreciate the reviewer's feedback and understand the need for a more in-depth explanation regarding the *Ucn3-Cre;Itgb1^f/f^* (*Itgb1^ΔMatureβ/ΔMatureβ^*) mice. Our hypothesis suggests that the Itgb1 deletion mediated by *Ngn3-Cre* occurs during early embryonic stages, indicating a key role for Itgb1 signaling in the migration and aggregation of progenitor cells in the endocrine lineage during islet development. This is supported by the absence of aggregation phenotypes in previous studies using the insulin promoter as a Cre driver for Itgb1 ablation.

To further explore whether Itgb1 signaling continues to maintain islet cell aggregation following β-cell maturation and islet formation, we implemented a *Ucn3-Cre*-mediated deletion of Itgb1, creating the *Itgb1^ΔMatureβ/ΔMatureβ^* model. Our choice of *Ucn3-Cre* is deliberate as Ucn3 is a late β-cell-specific maturity marker in mouse islets, initiating its expression perinatally and becoming prevalent by 3 weeks of age (as per (Blum et al., 2014; van der Meulen and Huising, 2014)). In our revised manuscript, we provide a more detailed explanation of our rationale for using *Ucn3-Cre*. Interestingly, the *Itgb1^ΔMatureβ/ΔMatureβ^* mice showed no observable abnormalities in blood glucose, aligning with earlier studies that used *RIP-Cre* to inactivate Itgb1 solely in β-cells. In addition, we conducted extensive studies on insulin secretion from the *Itgb1^ΔMatureβ/ΔMatureβ^* islets and found no significant differences between the control and mutant littermates. The consistency between our findings and those of previous studies using the *RIP-Cre; Itgb1* deletion model underscores the robustness of the *Itgb1^ΔMatureβ/ΔMatureβ^* phenotype. It provides strong support to our hypothesis that Itgb1-dependent islet aggregation is primarily regulated during early developmental stages, and Itgb1 signaling doesn't seem to be critical for establishing islet aggregation once the β-cells have fully matured.

In response to the reviewer's comments, we have included a statistical analysis of the relative β-cell population in our revised manuscript. This additional data corroborates our findings and bolsters our argument. Given the striking similarities between the *Itgb1^ΔMatureβ/ΔMatureβ^* and the *RIP-Cre; Itgb1* deletion models, as well as the robustness of our current and new data, we did not deem it necessary to conduct additional experiments on the *Itgb1^ΔMatureβ/ΔMatureβ^* islets. We believe our current evidence, supplemented by the statistical analysis and in alignment with prior studies, presents a compelling case that should address the reviewer's concerns effectively.

The RNA-seq data, although interesting, are only superficially analyzed, nor validated. Experiments of immunofluorescence do not really validate the most interesting alterations of genes found affected by the gene deletions, since it is not very quantitative. It would have been helpful to confirm at the protein level by Western blotting the expression of some of the most interesting genes, or at least by qPCR. The authors would be surprised to discover that RNA-seq does not necessarily translate into the same order of magnitude of mRNA levels when testing the same samples by qPCR.

We agree that immunofluorescence, while providing valuable visual and spatial information, is not the most quantitative method for validating gene expression changes. We also agree that RNA-seq data does not always correlate precisely with mRNA levels as determined by other methods like qPCR.

In our study, we have considered the immunofluorescence technique as 'semi-quantitative', referring to the careful methodological approach we have adopted. This method allows us to observe relative expression changes of key genes essential for endocrine cell maturation, which align with our RNA-seq results. While it may not be possible to validate every differentially expressed gene due to experimental constraints, this method gives us a reasonable estimate of their behavior. In response to the reviewer’s suggestion, we have included qPCR analysis in our revised manuscript to offer a more quantitative measure of gene expression changes. Due to the challenge of isolating a sufficient quantity of islets from mutant mice, Western blotting was not feasible in this context. Thus, we used qPCR analysis to study relative gene expression between controls and both *Itgb1^ΔEndo/ΔEndo^* (Figure S2A) and *Ctnna1^ΔEndo/ΔEndo^* (Figure S6A) islets We selected a set of up-regulated and down-regulated genes (total of 10), all of which are crucial for endocrine cell lineage and function.

Importantly, our qPCR results align well with our RNA-seq findings, further supporting the observed defects in endocrine cell lineages and functional gene expression in mutant islets. We hope that these additional data and clarifications address the reviewer's concerns and provide a more comprehensive view of our work.

Reviewer #2 (Recommendations for the authors):The goal of this study was to uncover mechanisms that govern the aggregation and "self-assembly" of islets during embryonic development and postnatally, a process that is not well understood. The authors focused on the role of integrin-β 1 (Itgb1), a key mediator of cell-extracellular matrix (ECM) interactions, and on α-catenin (Ctnna1), a hub protein that links cell adhesion molecules with the cytoskeleton. They used conditional knockout models (Ngn3-Cre and Ucn3-Cre), microscopy and morphometric image analysis, and transcriptomic, and functional measures of glucose metabolism. They found that loss of Itgb1 from endocrine precursor cells resulted in premature aggregation of islets and impaired migration of islets from pancreatic ducts resulting in disturbed islet morphology. Additionally, they observed loss of islet vascularization and decreased expression of markers of endocrine cell identity, which together resulted in glucose intolerance. On the other hand, loss of Ctnna1 resulted in smaller, hypervascularized islets which also expressed fewer markers of differentiated endocrine cell identity resulting in an age-dependent loss of glucose tolerance. Ctnna1 mice also exhibited loss of glucagon and somatostatin-positive islets cells, with loss of glucagon+ cells occurring postnatally. The authors concluded from these studies that a balance exists between cell-ECM and cell-cell interactions, and that loss of one of these interactions (as occurs in Itgb1 and Ctnna1 deficient mice, respectively) may result in an increase in the other. However, the loss of both Itgb1 and Ctnna1 from endocrine precursor cells resulted in an almost complete loss of islet aggregation and severe glucose intolerance.The conclusions of this paper are mostly supported by the data. Strengths include the thorough characterization of mice, especially the Itgb1 mice with the Ngn3-Cre driver. The work shows a clear effect of both Itgb1 and Ctnna1 to modulate islet aggregation/morphology without affecting cell specification or overall abundance (during development). The study provides compelling evidence for the role of these cell adhesion molecules in governing islet aggregation, morphology, identity, and function and is impactful for the field.

We are grateful for the reviewer’s detailed review and recognition of the significance of our research. We appreciate her feedback emphasizing the value of our findings on the roles of Itgb1 and Ctnna1 in islet aggregation, morphology, identity, and function, and the impact of this work on the islet biology field.

In response to these comments, and similar feedback from other reviewers, we have made significant revisions in our resubmission. To address the need for more comprehensive statistical evidence, we now provide thorough statistical analyses, including all pertinent numerical values, in Supplementary files, entitled “List of Replicates and Statistical Analysis” and “Source Data spreadsheet”.

We agree with the reviewer's assertion that other mediators of cell-ECM interactions may play crucial roles in islet aggregation. While Itgb1 is pivotal for these interactions, compensatory mechanisms involving other integrins, such as beta3, beta4, beta5, beta6, and alpha3, alpha6, and alphaV integrins (Cirulli et al., 2000; Krishnamurthy et al., 2011; Schiesser et al., 2021; Yashpal et al., 2005), might maintain cell-ECM interaction in Itgb1-deficient islets and be essential for islet survival and function. We have incorporated these points into the discussion in our resubmitted manuscript.

We have also revised our figures with high-resolution Airyscan images, providing a clearer depiction of cell shape alterations and cell-cell adhesion in *Itgb1* mutant islets. Furthermore, we have now included evidence of an increased number of TUNEL^+^ apoptotic glucagon-expressing α cells in *Ctnna1* mutant islets (Figure 4L-4N), which further bolsters our conclusion. We sincerely appreciate the reviewer's insightful comments, which have strengthened our manuscript.

Reviewer #3 (Recommendations for the authors):Pancreatic islet morphogenesis is a complex interplay between endocrine cell-intrinsic signaling coupled with external cues from ECM and neighboring cells. Defining the specific pathways and the developmental window when they are critical to the eventual islet architecture and function still remains unclear. This study adds to the body of evidence that the ECM-Integrin signaling and cell-cell adhesion signaling mediated via α-catenin has a critical combinatorial role in determining the islet cell aggregation, vascularization, and eventual function.Strengths:The reciprocal regulation of the ECM-integrin pathway and cell-cell adhesion via the α-catenin pathway identified in this study provides insight into how the external cues for developing endocrine cells are processed and integrated to eventually allow mouse islets to develop a core of β-cells and the mantle comprising of other endocrine cells. The detailed dissection to identify the developmental window using in vivo studies complemented with explant studies with lineage tracing is compelling strength of this study.The predicted changes in mouse islet (mantle-core) architecture due to changes in cell-cell adhesion strength as validated by the combination of the models that the authors use in this study provides a better understanding of the dynamics of islet morphogenesis.Weakness:The authors suggest that hypoxia resulting from changes in the vasculature is a potential cause of functional immaturity stemming from the expression of disallowed genes and enrichment of genes in RNA-seq involved in Hypoxic responses. Experimental demonstration of hypoxia and its causative role in these models is however lacking to substantiate this explanation.Many other groups have previously demonstrated the importance of α-catenin and β-integrin 1 in islet morphogenesis and function, using earlier (Pdx1 promoter-driven Cre) and later (various insulin promoter-driven Cre) developmental windows. Though this study narrows down the window to when the endocrine lineage is determined by Ngn3 and suggests that the altered cell adhesion leads to changes in function and aggregation, intracellular signaling pathways suggested by other studies, such as the sonic hedgehog pathway, PARP, and others remain unexplored in the current context to provide more mechanistic insight.While the regulation of the vascularization of the islets in both these mutant models (Itfb1 and Ctnna1) individually and in combination is very convincing, the authors note that this was with no change in angiogenic factor expression. While their explanation invoking the alterations of cortical tension by changes in actin cytoskeleton is certainly plausible and intriguing, it is not demonstrated experimentally and remains to be established in future work.

We greatly appreciate the insightful feedback from the reviewer, and we value the recognition of the strengths of our study. Regarding the critiques, we acknowledge the gaps in our study and are thankful for the constructive suggestions that have helped us to improve our work.

Firstly, we concur that our current study does not provide direct experimental validation of hypoxia as the cause for functional immaturity in *Itgb1* and *Ctnna1* mutant models. This interpretation is speculative, stemming from observed changes in vasculature and gene enrichment in hypoxic responses. To better reflect this point, we have revised our discussion to clearly state that the involvement of hypoxia is a hypothesis based on these observations and requires further investigation.

Secondly, we acknowledge the important role of Ctnna1 and Itgb1 in islet morphogenesis, as demonstrated by previous research using *Pdx1*-promoter-driven Cre and insulin promoter-driven Cre. Our study contributes to these findings by narrowing down the developmental window to the point at which the endocrine lineage is determined by Ngn3. While we recognize the potential relevance of other intracellular signaling pathways, such as the sonic hedgehog pathway and PARP, our RNA-seq data did not indicate significant changes in these pathways in our models. We have, however, included in our discussion the possibility of other intracellular signaling pathways playing a role, and we aim to further investigate these in future studies.

Finally, we agree with the reviewer that our proposed mechanism involving alterations in cortical tension due to changes in the actin cytoskeleton remains speculative. While the involvement of actin cytoskeleton alterations in vascularization is an intriguing possibility, we consider it beyond the scope of the current work. We have included this speculation in the discussion.

Once again, we thank the reviewer for his constructive feedback. His comments not only enhance the quality of our present work but also help guide our future research directions.

1. Concurrent measurement of insulin and glucose during the GTT provides compelling evidence of a β-cell defect. Adding this with the GTT data in Figures 1, 4, and S6 would add to the strength of the study.

We acknowledge the significance of concurrent measurements of insulin and glucose during the Glucose Tolerance Test (GTT) and appreciate the reviewer's suggestion. In line with this, we have incorporated GTT data for the mutant mice and depicted defective insulin secretion during glucose challenge in *Itgb1^∆endo/∆endo^* (Figure 1K) and *Ctnna1; Itgb1^∆endo/∆endo^* (Figure 7J) mice.

We observed mild glucose intolerance in *Ctnna1^∆endo/∆endo^* mice at 8 weeks (Figure 4B); however, insulin content quantification in serum after glucose challenge did not demonstrate any substantial defect in insulin secretion for the *Ctnna1^∆endo/∆endo^* mice (Figure 4C). Given the mild glucose intolerance in the *Ctnna1^∆endo/∆endo^* mice, we considered the possibility that conventional serum insulin measurements might not adequately represent alterations in β-cell function. To better understand this, we employed a more sensitive detection method, the hyperglycemic clamp technique, to quantify in vivo insulin secretion. Employing this assay, we noticed a trend toward diminished overall insulin secretion in the *Ctnna1^∆endo/∆endo^* mice at 8 weeks (Figure S6B,S6C), supplementing our evidence that Ctnna1-dependent islet cell aggregation influences β-cell function.

2. The glucose and functional phenotype of the Ucn-Cre mediated Itgb1ΔMatureβ/ΔMatureβ has not been described. Since this is a new model generated by the authors, it is important to describe the phenotype and also compare and contrast it to the other previously described mature β-cell ITGB knockouts that the authors allude to in their discussion.

We appreciate the reviewer's insightful comment on the *Ucn3-Cre* mediated *Itgb1^ΔMatureβ/ΔMatureβ^* model. As previously explained, the *Ucn3-Cre* driver follows the expression of Urocortin 3 (Ucn3), a specific marker of mature β-cell. Its expression begins during the perinatal period and becomes widespread around 3 weeks of age, as documented by van der Meulen et al., 2014, and Blum et al., 2014. Consequently, we utilized *Ucn3-Cre* for *Itgb1* deletion in our *Itgb1^ΔMatureβ/ΔMatureβ^* model, enabling us to investigate the role of Itgb1 signaling in preserving islet cell aggregation following β-cell maturation and islet formation.

In the revised manuscript, we've clarified this model choice and its implications by providing additional context in both the results and Discussion sections. Notably, we have included that our *Itgb1^ΔMatureβ/ΔMatureβ^* mice exhibit no detectable metabolic issues. We also performed comprehensive metabolic analyses such as calcium imaging and found no observable functional phenotype. We have not included these data in the current manuscript as they go beyond the scope of this study. If the reviewer deems it necessary, we are more than willing to incorporate these findings in the revised version.

As for the lack of observable glucose and functional phenotype in the *Itgb1^ΔMatureβ/ΔMatureβ^* model, we find it consistent with what we might expect. Given that the *RIP-Cre* mediated *Itgb1* deletion encompasses a large proportion of the Ucn3^+^ β-cells, it's reasonable that the *Itgb1^ΔMatureβ/ΔMatureβ^* mice wouldn't show more pronounced phenotypes than the *RIP-Cre* mediated *Itgb1* deletion mice. We hope this explanation provides better understanding of our approach.

3. Page 6, Line 219 – Should be 'Figure 2K, 2L' and not 'Figure 1K, 1l.'

We appreciate the reviewer's careful eye for detail. We have corrected the figure references in the revised manuscript. Thank you for bringing this oversight to our attention.